# Provably Fast Convergence of Independent Natural Policy Gradient for Markov Potential Games

**Youbang Sun**[*]
Northeastern University
sun.youb@northeastern.edu

**Tao Liu**[*]
Texas A&M University
tliu@tamu.edu

**Ruida Zhou**
Texas A&M University
ruida@tamu.edu

**P. R. Kumar**
Texas A&M University
prk@tamu.edu

**Shahin Shahrampour**
Northeastern University
s.shahrampour@northeastern.edu

## Abstract

This work studies an independent natural policy gradient (NPG) algorithm for the multi-agent reinforcement learning problem in Markov potential games. It is shown that, under mild technical assumptions and the introduction of the *suboptimality gap*, the independent NPG method with an oracle providing exact policy evaluation asymptotically reaches an $\epsilon$-Nash Equilibrium (NE) within $\mathcal{O}(1/\epsilon)$ iterations. This improves upon the previous best result of $\mathcal{O}(1/\epsilon^2)$ iterations and is of the same order, $\mathcal{O}(1/\epsilon)$, that is achievable for the single-agent case. Empirical results for a synthetic potential game and a congestion game are presented to verify the theoretical bounds.

## 1 Introduction

Reinforcement learning (RL) is often impacted by the presence and interactions of several agents in a multi-agent system. This challenge has motivated recent studies of multi-agent reinforcement learning (MARL) in stochastic games [37, 3]. Applications of MARL include robotics [30], modern production systems [2], economic decision making [33], and autonomous driving [31]. Among the various types of stochastic games, we focus on a commonly studied model for MARL, known as Markov Potential Games (MPGs). MPGs are seen as a generalization of canonical Markov Decision Processes (MDPs) in the multi-agent setting. In MPGs, there exists a potential function that can track the value changes of all agents. Unlike single-agent systems, where the goal is to find the optimal policy, the objective in this paper is to find a global policy, formed by the joint product of a set of local policies, that leads the system to reach a Nash equilibrium (NE) [29], which is precisely defined in Section 2.

A major challenge in the analysis of multi-agent systems is the restriction on joint policies of agents. For single-agent RL, policy updates are designed to increase the probability of selecting the action with the highest reward. However, in multi-agent systems, the global policy is constructed by taking the product of local agents' policies, which makes MARL algorithms suffer a greater risk of being trapped near undesirable stationary points. Consequently, finding a NE in MARL is more challenging than finding the global optimum in the single-agent case, and it is therefore difficult for MPGs to recover the convergence rates of single-agent Markov decision processes (MDPs).

Additionally, the global action space in MPGs scales exponentially with the number of agents within the system, making it crucial to find an algorithm that scales well for a large number of agents. Recent

---

[*]The first two authors contributed equally.

37th Conference on Neural Information Processing Systems (NeurIPS 2023).

Table 1: Convergence rate results for policy gradient-based methods in Markov potential games. Some results have been modified to ensure better comparison.

| ALGORITHM | ITERATION COMPLEXITY [2] |
|---|---|
| PG + DIRECT [39, 19] | $\mathcal{O}\left(\frac{\sum_{i=1}^{n}|\mathcal{A}_i|M^2}{(1-\gamma)^4\epsilon^2}\right)$ |
| PG + SOFTMAX [38] | $\mathcal{O}\left(\frac{n\max_i|\mathcal{A}_i|M^2}{(1-\gamma)^4c^2\epsilon^2}\right)$ |
| NPG + SOFTMAX [38] | $\mathcal{O}\left(\frac{nM}{(1-\gamma)^3c\epsilon^2}\right)$ |
| PG + SOFTMAX + LOG-BARRIER REG. [38] | $\mathcal{O}\left(\frac{n\max_i|\mathcal{A}_i|^2M^2}{(1-\gamma)^4\epsilon^2}\right)$ |
| NPG + SOFTMAX + LOG-BARRIER REG. [38] | $\mathcal{O}\left(\frac{n\max_i|\mathcal{A}_i|M^2}{(1-\gamma)^4\epsilon^2}\right)$ |
| PROJECTED Q ASCENT [8] | $\mathcal{O}\left(\frac{n^2\max_i|\mathcal{A}_i|^2M^2}{(1-\gamma)^7\epsilon^4}\right)$ OR $\mathcal{O}\left(\frac{n\max_i|\mathcal{A}_i|M^4}{(1-\gamma)^2\epsilon^2}\right)$ |
| PROJECTED Q ASCENT (FULLY COOP) [8] | $\mathcal{O}\left(\frac{n\max_i|\mathcal{A}_i|M}{(1-\gamma)^3\epsilon^2}\right)$ |
| (OURS) NPG + SOFTMAX | $\mathcal{O}\left(\frac{\sqrt{n}M}{(1-\gamma)^2c\delta^*\epsilon}\right)$ [3] |

studies [8, 14] addressed the issue by an approach called independent learning, where each agent performs policy update based on local information without regard to policies of the other agents. Independent learning algorithms scale only linearly with respect to the number of agents and are therefore preferred for large-scale multi-agent problems.

Using algorithms such as policy gradient (PG) and natural policy gradient (NPG), single-agent RL can provably converge to the global optimal policy [1]. However, extending these algorithms from single-agent to multi-agent settings presents natural challenges as discussed above. Multiple recent works have analyzed PG and NPG in multi-agent systems. However, due to the unique geometry of the problem and complex relationships among the agents, the theoretical understanding of MARL is still limited, with most works showing slower convergence rates when compared to their single-agent counterparts (see Table 1).

**Contributions** We study the independent NPG algorithm in multi-agent systems and provide a novel technical analysis that guarantees a provably fast convergence rate. We start our analysis with potential games in Section 3.1 and then generalize the findings and provide a convergence guarantee for Markov potential games in Section 3.2. We show that under mild assumptions, the ergodic (i.e., temporal average of) NE-gap converges with iteration complexity of $\mathcal{O}(1/\epsilon)$ after a finite threshold (Theorem 3.6). This result provides a substantial improvement over the best known rate of $\mathcal{O}(1/\epsilon^2)$ in [38]. Our main theorem also reveals mild or improved dependence on multiple critical factors, including the number of agents $n$, the initialization dependent factor $c$, the distribution mismatch coefficient $M$, and the discount factor $\gamma$, discussed in Section 3. We dedicate Section 3.3 to discuss the impact of the asymptotic suboptimality gap $\delta^*$, which is a new factor in this work.

In addition to our theoretical results, two numerical experiments are also conducted in Section 4 for verification of the analysis. We consider a synthetic potential game similar to [38] and a congestion game from [19]. The omitted proofs of our theoretical results can be found in the appendix.

## 1.1 Related Literature

**Markov Potential Games** Since many properties of single-agent RL do not hold in MARL, the analysis of MARL presents several challenges. Various settings have been addressed for MARL in recent works. A major distinction between these works stems from whether the agents are competitive or cooperative [11]. In this paper we consider MPGs introduced in stochastic control [7]. MPGs are a generalized formulation of identical-reward cooperative games. Markov cooperative games have been studied in the early work of [34], and more recently by [23, 24, 32, 10]. The work [36]

---

[2]$c$, $M$ and $\delta^*$ will be formally defined in Section 3. Note that [8] has a different definition of $M$. For a fair comparison, we convert the results of [8] into the current definition of $M$.

[3]This iteration complexity holds after a finite threshold.

also offered extensive empirical results for cooperative games using multi-agent proximal policy optimization. The work by [27] established polynomial convergence for MPGs under both Q-update as well as actor-critic update. [25] studied an independent Q-update in MPGs with perturbation, which converges to a stationary point with probability one. Conversely, [17, 13] studied potential games, a special static case of simplified MPGs with no state transition.

**Policy Gradient in Games**  Policy gradient methods for centralized MDPs have drawn much attention thanks to recent advancements in RL theory [1, 26]. The extension of PG methods to multi-agent settings is quite natural. [6, 35, 5] studied two-player zero-sum competitive games. The general-sum linear-quadratic game was studied in [12]. [14] studied general-sum Markov games and provided convergence of V-learning in two-player zero-sum games.

Of particular relevance to our work are the works [19, 39, 38, 8] which focus on the MPG setting and propose adaptations of PG and NPG-based methods from single-agent problems to the MARL setting. Table 1 provides a detailed comparison between these works. The previous theoretical results in multi-agent systems have provided convergence rates dependent on different parameters of the system. However, the best-known iteration complexity to reach an $\epsilon$-NE in MARL is $\mathcal{O}(1/\epsilon^2)$. Therefore, there still exists a rate discrepancy between MARL methods where $\mathcal{O}(1/\epsilon)$ complexity has been established in centralized RL algorithms [1]. Our main contribution is to close this gap by establishing an iteration complexity of $\mathcal{O}(1/\epsilon)$ in this work.

## 2  Problem Formulation

We consider a stochastic game $\mathcal{M} = (n, \mathcal{S}, \mathcal{A}, P, \{r_i\}_{i \in [n]}, \gamma, \rho)$ consisting of $n$ agents denoted by a set $[n] = \{1, ..., n\}$. The global action space $\mathcal{A} = \mathcal{A}_1 \times ... \times \mathcal{A}_n$ is the product of individual action spaces, with the global action defined as $\boldsymbol{a} := (a_1, ..., a_n)$. The global state space is represented by $\mathcal{S}$, and the system transition model is captured by $P : \mathcal{S} \times \mathcal{A} \to \Delta(\mathcal{S})$. Furthermore, each agent is equipped with an individual reward function $r_i : \mathcal{S} \times \mathcal{A} \to [0, 1]$. We use $\gamma \in (0, 1)$ to denote the discount factor and $\rho \in \Delta(\mathcal{S})$ to denote the initial state distribution.

The system policy is denoted by $\pi : \mathcal{S} \to \Delta(\mathcal{A}_1) \times \cdots \times \Delta(\mathcal{A}_n) \subset \Delta(\mathcal{A})$, where $\Delta(\mathcal{A})$ is the probability simplex over the global action space. In the multi-agent setting, all agents make decisions independently given the observed state, often referred to as a *decentralized* stochastic policy [38]. Under this setup, we have $\pi(\boldsymbol{a}|s) = \prod_{i \in [n]} \pi_i(a_i|s)$, where $\pi_i : \mathcal{S} \to \Delta(\mathcal{A}_i)$ is the local policy for agent $i$. For the ease of notation, we denote the joint policy over the set $[n] \backslash \{i\}$ by $\pi_{-i} = \prod_{j \in [n] \backslash \{i\}} \pi_j$ and use the notation $a_{-i}$ analogously.

We define the state value function $V_i^\pi(s)$ with respect to the reward $r_i(s, \boldsymbol{a})$ as $V_i^\pi(s) := \mathbb{E}^\pi[\sum_{t=0}^\infty \gamma^t r_i(s^t, \boldsymbol{a}^t)|s^0 = s]$, where $(s^t, \boldsymbol{a}^t)$ denotes the global state-action pair at time $t$, and we denote the expected value of the state value function over the initial state distribution $\rho$ as $V_i^\pi(\rho) := \mathbb{E}_{s \sim \rho}[V_i^\pi(s)]$. We can similarly define the state visitation distribution under $\rho$ as $d_\rho^\pi(s) := (1 - \gamma)\mathbb{E}^\pi[\sum_{t=0}^\infty \gamma^t \mathbb{1}(s_t = s)|s_0 \sim \rho]$, where $\mathbb{1}$ is the indicator function. The state-action value function and advantage function are, respectively, given by

$$Q_i^\pi(s, \boldsymbol{a}) = \mathbb{E}^\pi[\sum_{t=0}^\infty \gamma^t r_i(s^t, \boldsymbol{a}^t)|s^0 = s, \boldsymbol{a}^0 = \boldsymbol{a}], \quad A_i^\pi(s, \boldsymbol{a}) = Q_i^\pi(s, \boldsymbol{a}) - V_i^\pi(s). \quad (1)$$

For the sake of analysis, we further define the marginalized Q-function and advantage function $\bar{Q}_i : \mathcal{S} \times \mathcal{A}_i \to \mathbb{R}$ and $\bar{A}_i : \mathcal{S} \times \mathcal{A}_i \to \mathbb{R}$ as:

$$\bar{Q}_i^\pi(s, a_i) := \sum_{a_{-i}} \pi_{-i}(a_{-i}|s) Q_i^\pi(s, a_i, a_{-i}), \quad \bar{A}_i^\pi(s, a_i) := \sum_{a_{-i}} \pi_{-i}(a_{-i}|s) A_i^\pi(s, a_i, a_{-i}). \quad (2)$$

**Definition 2.1** ([39])**.** *The stochastic game $\mathcal{M}$ is a Markov potential game if there exists a bounded potential function $\phi : \mathcal{S} \times \mathcal{A} \to \mathbb{R}$ such that for any agent $i$, initial state $s$ and any set of policies $\pi_i, \pi_i', \pi_{-i}$:*

$$V_i^{\pi_i', \pi_{-i}}(s) - V_i^{\pi_i, \pi_{-i}}(s) = \Phi^{\pi_i', \pi_{-i}}(s) - \Phi^{\pi_i, \pi_{-i}}(s),$$

*where $\Phi^\pi(s) := \mathbb{E}^\pi[\sum_{k=0}^\infty \gamma^k \phi(s^k, \boldsymbol{a}^k)|s^0 = s]$.*

We assume that an upper bound exists for the potential function, i.e., $0 \leq \phi(s, \boldsymbol{a}) \leq \phi_{max}, \forall s \in \mathcal{S}, \boldsymbol{a} \in \mathcal{A}$, and consequently, $\Phi^\pi(s) \leq \frac{\phi_{max}}{1-\gamma}$.

It is common in policy optimization to parameterize the policy for easier computations. In this paper, we focus on the widely used softmax parameterization [1, 26], where a global policy $\pi(\boldsymbol{a}|s) = \prod_{i \in [n]} \pi_i(a_i|s)$ is parameterized by a set of parameters $\{\theta_1, ..., \theta_n\}, \theta_i \in \mathbb{R}^{|\mathcal{S}| \times |\mathcal{A}_i|}$ in the following form

$$\pi_i(a_i|s) = \frac{\exp\{[\theta_i]_{s,a_i}\}}{\sum_{a_j \in \mathcal{A}_i} \exp\{[\theta_i]_{s,a_j}\}}, \ \forall(s, a_i) \in \mathcal{S} \times \mathcal{A}_i.$$

### 2.1 Optimality Conditions

In the MPG setting, there may exist multiple stationary points, a set of policies that has zero policy gradients, for the same problem; therefore, we need to introduce notions of solutions to evaluate policies. The term *Nash equilibrium* is used to define a measure of "stationarity" in strategic games.

**Definition 2.2.** *A joint policy $\pi^*$ is called a Nash equilibrium if for all $i \in [n]$, we have*

$$V_i^{\pi_i^*, \pi_{-i}^*}(\rho) \geq V_i^{\pi_i', \pi_{-i}^*}(\rho) \ \ \text{for all } \pi_i'.$$

For any given joint policy that does not necessarily satisfy the definition of NE, we provide the definition of NE-gap as follows [39]:

$$\text{NE-gap}(\pi) := \max_{i \in [n], \pi_i' \in \Delta(\mathcal{A}_i)} \left[ V_i^{\pi_i', \pi_{-i}}(\rho) - V_i^{\pi_i, \pi_{-i}}(\rho) \right].$$

Furthermore, we refer to a joint policy $\pi$ as $\epsilon$-NE when its NE-gap$(\pi) \leq \epsilon$. The NE-gap satisfies the following inequalities based on the performance difference lemma [15, 38],

$$\text{NE-gap}(\pi) \leq \frac{1}{1-\gamma} \sum_{i,s,a_i} d_\rho^{\pi_i^*, \pi_{-i}}(s) \pi_i^*(a_i|s) \bar{A}_i^\pi(s, a_i) \leq \frac{1}{1-\gamma} \sum_{i,s} d_\rho^{\pi_i^*, \pi_{-i}}(s) \max_{a_i} \bar{A}_i^\pi(s, a_i).$$

In the tabular single-agent RL, most works consider the optimality gap as the difference between the expectations of the value functions of the current policy and the optimal policy, defined as $V^{\pi^k}(\rho) - V^{\pi^*}(\rho)$. However, this notion does not extend to multi-agent systems. Even in a fully cooperative MPG where all agents share the same reward, the optimal policy of one agent is dependent on the joint policies of other agents. As a result, it is common for the system to have multiple "best" policy combinations (or stationary points), which all constitute Nash equilibria. Additionally, it has also been addressed by previous works that any NE point in an MPG is first order stable [39]. Given that this work addresses a MARL problem, we focus our analysis on the NE-gap.

## 3 Main Results

### 3.1 Warm-Up: Potential Games

In this section, we first consider the instructive case of static potential games, where the state does not change with time. Potential games are an important class of games that admit a potential function $\phi$ to capture differences in each agent's reward function caused by unilateral bias [28, 13], which is defined as

$$r_i(a_i, a_{-i}) - r_i(a_i', a_{-i}) = \phi(a_i, a_{-i}) - \phi(a_i', a_{-i}), \quad \forall a_i, a_i', a_{-i}. \tag{3}$$

**Algorithm Update** In the potential games setting, the policy update using natural policy gradient is [4]:

$$\pi_i^{k+1}(a_i) \propto \pi_i^k(a_i) \exp\left(\eta \bar{r}_i^k(a_i)\right), \tag{4}$$

where the exact independent gradient over policy $\pi_i$, also referred to as oracle, is captured by the marginalized reward $\bar{r}_i(a_i) = \mathbb{E}_{a_{-i} \sim \pi_{-i}}[r_i(a_i, a_{-i})]$. By definition, the NE-gap for potential games

is calculated as $\max_{i \in [n]} \langle \pi_i^{*k} - \pi_i^k, \bar{r}_i^k \rangle$, where $\pi_i^{*k} \in \arg\max_{\pi_i} V_i^{\pi_i, \pi_{-i}^k}$ is the optimal solution for agent $i$ when the rest of the agents use the joint policy $\pi_{-i}^k$.

The local marginalized reward $\bar{r}_i(a_i)$ is calculated based on other agents' policies; hence, for any two sets of policies, the difference in marginalized reward can be bounded using the total variation distance of the two probability measures [4]. Using this property, we can also show that there is a "smooth" relationship between the marginalized rewards and their respective policies. We note that this relationship holds for stochastic games in general. It does not depend on the nature of the policy update or the potential game assumption.

We now introduce a lemma that is specific to the potential game formulation and the NPG update:

**Lemma 3.1.** *Given policy $\pi^k$ and marginalized reward $\bar{r}_i^k(a_i)$, for $\pi^{k+1}$ generated using an NPG update in (4), we have the following inequality for any $\eta < \frac{1}{\sqrt{n}}$,*

$$\phi(\pi^{k+1}) - \phi(\pi^k) \geq (1 - \sqrt{n}\eta) \sum_{i=1}^n \langle \pi_i^{k+1} - \pi_i^k, \bar{r}_i^k \rangle.$$

Lemma 3.1 provides a lower bound on the difference of potential functions between two consecutive steps. This implies that at each time step, the potential function value is guaranteed to be monotonically increasing, as long as the learning rate satisfies $\eta < \frac{1}{\sqrt{n}}$.

Note that the lower bound of Lemma 3.1 involves $\langle \pi_i^{k+1} - \pi_i^k, \bar{r}_i^k \rangle$, which resembles the form of NE-gap $\langle \pi_i^{*k} - \pi_i^k, \bar{r}_i^k \rangle$. Assuming we can establish a lower bound for the right-hand side of Lemma 3.1 using NE-gap, the next step is to show that the sum of all NE-gap iterations is upper bounded by a telescoping sum of the potential function, thus obtaining an upper bound on the NE-gap.

We start by introducing a function $f^k : \mathbb{R} \to \mathbb{R}$ defined as follows,

$$f^k(\alpha) = \sum_{i=1}^n \langle \pi_{i,\alpha} - \pi_i^k, \bar{r}_i^k \rangle, \text{ where } \pi_{i,\alpha}(\cdot) \propto \pi_i^k(\cdot) \exp\{\alpha \bar{r}_i^k(\cdot)\}. \tag{5}$$

It is obvious that $f^k(0) = 0$, $f^k(\eta) = \sum_i \langle \pi_i^{k+1} - \pi_i^k, \bar{r}_i^k \rangle \geq 0$, and $\lim_{\alpha \to \infty} f^k(\alpha) = \sum_i \langle \pi_i^{*k} - \pi_i^k, \bar{r}_i^k \rangle$.

Without loss of generality, for agent $i$ at iteration $k$, define $a_{i_p}^k \in \arg\max_{a_j \in \mathcal{A}_i} \bar{r}_i^k(a_j) =: \mathcal{A}_{i_p}^k$ and $a_{i_q}^k \in \arg\max_{a_j \in \mathcal{A}_i \setminus \mathcal{A}_{i_p}^k} \bar{r}_i^k(a_j)$, where $\mathcal{A}_{i_p}^k$ denotes the set of the best possible actions for agent $i$ at iteration $k$. Similar to [38], we define

$$c^k := \min_{i \in [n]} \sum_{a_j \in \mathcal{A}_{i_p}^k} \pi_i^k(a_j) \in (0, 1), \quad \delta^k := \min_{i \in [n]} [\bar{r}_i^k(a_{i_p}^k) - \bar{r}_i^k(a_{i_q}^k)] \in (0, 1). \tag{6}$$

Additionally, we denote $c_K := \min_{k \in [K]} c^k$; $c := \inf_K c_K > 0$; $\delta_K := \min_{k \in [K]} \delta^k$.

We provide the following lemma on the relationship between $f^k(\alpha)$ and $f^k(\infty) := \lim_{\alpha \to \infty} f^k(\alpha)$, which lays the foundation to obtain sharper results than those in the existing work.

**Lemma 3.2.** *For function $f^k(\alpha)$ defined in (5) and any $\alpha > 0$, we have the following inequality.*

$$f^k(\alpha) \geq f^k(\infty) \left[ 1 - \frac{1}{c(\exp(\alpha \delta_K) - 1) + 1} \right].$$

We refer to $\delta_K$ as the minimal suboptimality gap of the system for the first $K$ iterations, the effect of which will be discussed later in Section 3.3. Using the two lemmas above and the definitions of $c$ and $\delta_K$, we establish the following theorem on the convergence of the NPG algorithm in potential games.

**Theorem 3.3.** *Consider a potential game with NPG update using (4). For any $K \geq 1$, choosing $\eta = \frac{1}{2\sqrt{n}}$, we have*

$$\frac{1}{K} \sum_{k=0}^{K-1} \text{NE-gap}(\pi^k) \leq \frac{2\phi_{max}}{K} \left( 1 + \frac{2\sqrt{n}}{c\delta_K} \right).$$

*Proof.* Choose $\alpha = \eta = \frac{1}{2\sqrt{n}}$ in Lemma 3.2,

$$\text{NE-gap}(\pi^k) \le \lim_{\alpha \to \infty} f^k(\alpha) \le \frac{1}{1 - \frac{1}{c(\exp(\delta_K \eta)-1)+1}} f^k(\eta) \le \frac{1}{1 - \frac{1}{c\delta_K \eta + 1}} f^k(\eta)$$

$$\le \frac{1}{(1 - \sqrt{n}\eta)\frac{c\delta_K \eta}{c\delta_K \eta + 1}}[\phi(\pi^{k+1}) - \phi(\pi^k)] = 2[\phi(\pi^{k+1}) - \phi(\pi^k)](1 + \frac{2\sqrt{n}}{c\delta_K}).$$

Then, we have $\frac{1}{K}\sum_{k=0}^{K-1} \text{NE-gap}(\pi^k) \le \frac{2(\phi(\pi^K)-\phi(\pi^0))}{K}(1 + \frac{2\sqrt{n}}{c\delta_K}) \le \frac{2\phi_{max}}{K}(1 + \frac{2\sqrt{n}}{c\delta_K})$. $\qquad\square$

One challenge in MARL is that the NE-gap is not monotonic, so we must seek ergodic convergence results, which characterize the behavior of the temporal average of the NE-gap. Theorem 3.3 shows that for potential games with NPG policy update, the ergodic NE-gap of the system converges to zero with a rate $\mathcal{O}(1/(K\delta_K))$. When $\delta_K$ is uniformly lower bounded, Theorem 3.3 provides a significant speed up compared to previous convergence results. Apart from the iteration complexity, the NE-gap is also dependent linearly on $1/c$ and $1/\delta_K$. We address the effect of $c$ in the analysis here and defer the discussion of $\delta_K$ to Section 3.3. Under some mild assumptions, we can show that the system converges with a rate of $\mathcal{O}(1/K)$.

**The Effect of $c$** The convergence rate given by Theorem 3.3 scales with $1/c$, where $c$ might potentially be arbitrarily small. A small value for $c$ generally describes a policy that is stuck at some regions far from a NE, yet the policy gradient is small. It has been shown in [20] that these ill-conditioned problems could take exponential time to solve even in single-agent settings for policy gradient methods. The same issue also occurs to NPG in MARL, since the local Fisher information matrix can not cancel the occupancy measure and action probability of other agents. A similar problem has also been reported in the analysis of [38, 26] for the MPG setting. [38] proposed the addition of a log-barrier regularization to mitigate this issue. However, that comes at the cost of an $\mathcal{O}(1/(\lambda K))$ convergence rate to a $\lambda$-neighborhood solution, which is only reduced to the exact convergence rate of $\mathcal{O}(1/\sqrt{K})$ when $\lambda = 1/\sqrt{K}$. Therefore, this limitation may not be effectively avoided without impacting the convergence rate.

## 3.2 General Markov Potential Games

We now extend our analysis to MPGs. The analysis mainly follows a similar framework as potential games. However, the introduction of state transitions and the discount factor $\gamma$ add an additional layer of complexity to the problem, making it far from trivial. As pointed out in [19], we can construct MDPs that are potential games for every state, yet the entire system is not a MPG. Thus, the analysis of potential games does not directly apply to MPGs.

We first provide the independent NPG update for MPGs.

**Algorithm Update** For MPGs at iteration $k$, the independent NPG updates the policy as follows [38]:

$$\pi_i^{k+1}(a_i|s) \propto \pi_i^k(a_i|s) \exp\left(\frac{\eta \bar{A}_i^{\pi^k}(s, a_i)}{1 - \gamma}\right), \tag{7}$$

where $\bar{A}_i$ is defined in (2).

Different agents in a MPG do not share reward functions in general, which makes it difficult to compare evaluations of gradients across agents. However, with the introduction of Lemma 3.4, we find that MPGs have similar properties as fully cooperative games with a shared reward function. This enables us to establish relationships between policy updates of all agents. We first define $h_i(s, \boldsymbol{a}) := r_i(s, \boldsymbol{a}) - \phi(s, \boldsymbol{a})$, which implies $V_i^\pi(s) = \Phi^\pi(s) + V_{h_i}^\pi(s)$.

**Lemma 3.4.** *Define $\bar{A}_{h_i}^\pi(s, a_i)$ with respect to $h_i$ similar to (2). We then have*

$$\sum_{a_i}(\pi_i'(a_i|s) - \pi_i(a_i|s))\bar{A}_{h_i}^\pi(s, a_i) = 0, \quad \forall s \in \mathcal{S}, i \in [n], \pi_i', \pi_i \in \Delta(\mathcal{A}_i).$$

Lemma 3.4 shows a unique property of function $h_i$, where the expectation of the marginalized advantage function over every local policy $\pi_i'$ yields the same effect. This property is directly associated with the MPG problem structure and is later used in Lemma 3.5. Next, we introduce the following assumption on the state visitation distribution, which is crucial and standard for studying the Markov dynamics of the system.

**Assumption 3.1** ([1, 38, 8]). *The Markov potential game $\mathcal{M}$ satisfies:* $\inf_\pi \min_s d_\rho^\pi(s) > 0$.

Similar to potential games, when the potential function $\phi$ of a MPG is bounded, the marginalized advantage function $\bar{A}_i$ for two policies can be bounded by the total variation between the policies.

Additionally, similar to Lemma 3.1, we present a lower bound in the following lemma for the potential function difference in two consecutive rounds.

**Lemma 3.5.** *Given policy $\pi^k$ and marginalized advantage function $\bar{A}_i^{\pi^k}(s, a_i)$, for $\pi^{k+1}$ generated using NPG update in* (7)*, we have the following inequality,*

$$\Phi^{\pi^{k+1}}(\rho) - \Phi^{\pi^k}(\rho) \geq \left( \frac{1}{1-\gamma} - \frac{\sqrt{n}\phi_{max}\eta}{(1-\gamma)^3} \right) \sum_s d_\rho^{\pi^{k+1}}(s) \sum_{i=1}^n \langle \pi_i^{k+1}(\cdot|s), \bar{A}_i^{\pi^k}(s, \cdot) \rangle.$$

Thus, using a function $f$ adapted from (5) as a connection, we are able to establish the convergence of NE-gap for MPGs in the following theorem.

**Theorem 3.6.** *Consider a MPG with isolated stationary points and the policy update following NPG update (7). For any $K \geq 1$, choosing $\eta = \frac{(1-\gamma)^2}{2\sqrt{n}\phi_{max}}$, we have*

$$\frac{1}{K} \sum_{k=0}^{K-1} NE\text{-}gap(\pi^k) \leq \frac{2M\phi_{max}}{K(1-\gamma)} \left(1 + \frac{2\sqrt{n}\phi_{max}}{c\delta_K(1-\gamma)}\right),$$

*where*

$$M := \sup_\pi \max_s \frac{1}{d_\rho^\pi(s)},$$

$$c := \inf_{i \in [n], s \in \mathcal{S}, k \geq 0} \left( \sum_{a_j \in \mathcal{A}_{i_p}^k} \pi_i^k(a_j|s) \right) \in (0,1),$$

$$\delta^k := \min_{i \in [n], s \in \mathcal{S}} \left[ \bar{A}_i^{\pi^k}(s, a_{i_p}^k) - \bar{A}_i^{\pi^k}(s, a_{i_q}^k) \right], \quad \text{and} \quad \delta_K = \min_{0 \leq k \leq K-1} \delta^k \in (0,1),$$

*similar to* (6)*.*

Here, *isolated* implies that no other stationary points exist in any sufficiently small open neighborhood of any stationary point. This convergence result is similar to that provided in Theorem 3.3 for potential games. We note that our theorem also applies to an alternate definition for MPGs in works such as [8], which we discuss in Appendix C. Compared to potential games, the major difference is the introduction of $M$, which measures the distribution mismatch in the system. Generally, Assumption 3.1 implies a finite value for $M$ in the MPG setup.

**Discussion and Comparison on Convergence Rate**  Compared to the iteration complexity of previous works listed in Table 1, the convergence rate in Theorem 3.6 presents multiple improvements. Most importantly, the theorem guarantees that the averaged NE-gap reaches $\epsilon$ in $\mathcal{O}(1/\epsilon)$ iterations, improving the best previously known result of $\mathcal{O}(1/\epsilon^2)$ in Table 1. Furthermore, this rate of convergence does not depend on the size of action space $|\mathcal{A}_i|$, and it has milder dependence on other system parameters, such as the distribution mismatch coefficient $M$ and $(1-\gamma)$. Note that many of the parameters above could be arbitrarily large in practice (e.g., $1-\gamma = 0.01$ in our congestion game experiment). Theorem 3.6 indicates a tighter convergence bound with respect to the discussed factors in general.

### 3.3 The Consideration of Suboptimality Gap

In Theorems 3.3 and 3.6, we established the ergodic convergence rates of NPG in potential game and Markov potential game settings, which depend on $1/\delta_K$. In these results, $\delta^k$ generally encapsulates

the difference between the gradients evaluated at the best and second best policies, and $\delta_K$ is a lower bound on $\delta^k$. We refer to $\delta^k$ as the *suboptimality gap* at iteration $k$. In our analysis, the suboptimality gap provides a vehicle to establish the improvement of the potential function in two consecutive steps. In particular, it enables us to draw a connection between $\Phi^{\pi^{k+1}} - \Phi^{\pi^k}$ and NE-gap($\pi^k$) using Lemma 3.2 and Lemma B.3 in the appendix. On the contrary, most of the existing work does not rely on this approach and generally studies the relationship between $\Phi^{\pi^{k+1}} - \Phi^{\pi^k}$ and the *squared* NE-gap($\pi^k$), which suffers a slower convergence rate of $\mathcal{O}(1/\sqrt{K})$ [38].

The analysis leveraging the suboptimality gap, though has not been adopted in the MARL studies, was considered in the single-agent scenario. Khodadadian et al. [16] proved asymptotic geometric convergence of single-agent RL with the introduction of *optimal advantage function gap* $\Delta^k$, which shares similar definition as the gap $\delta^k$ studied in this work. Moreover, the notion of suboptimality gap is commonly used in the multi-armed bandit literature [18], so as to give the instance-dependent analysis.

While our convergence rate is sharper, a lower bound on the suboptimality gap is generally not guaranteed, and scenarios with zero optimality gap can be constructed in MPGs. In practice, we find that even if $\delta^k$ *approaches* zero in certain iterations, it may not *converge* to zero, and in these scenarios the system still converges without a slowdown. We provide a numerical example in Section 4.1 (Fig. 1a) to support our claim. Nevertheless, in what follows, we further identify a sufficient condition that allows us to alleviate this problem altogether by focusing on the asymptotic profile of the sub-optimality gap.

**Theoretical Relaxation**   The following proposition guarantees the asymptotic results of independent NPG.

**Proposition 3.1** ([38]). *Suppose that Assumption 3.1 holds and that the stationary policies are isolated. Then independent NPG with $\eta \leq \frac{(1-\gamma)^2}{2\sqrt{n}\phi_{max}}$ guarantees that $\lim_{k\to\infty} \pi^k = \pi^\infty$, where $\pi^\infty$ is a Nash policy.*

Since asymptotic convergence of policy is guaranteed by Proposition 3.1, the suboptimality gap $\delta^k$ is also guaranteed to converge to some $\delta^*$. We make the following assumption about the asymptotic suboptimality gap, which only depends on the property of the game itself.

**Assumption 3.2.** *Assume that $\lim_{k\to\infty} \delta^k = \delta^* > 0$.*

Assumption 3.2 provides a relaxation in the sense that instead of requiring a lower bound $\inf \delta^k$ for all $k$, we only need $\delta^* > 0$, a lower bound on the limit as the agents approach some Nash policies. This will allow us to disregard the transition behavior of the system and focus on the rate for large enough $k$.

By the definition of $\delta^*$, we know that there exists finite $K'$ such that $\forall k > K', |\delta^k - \delta^*| \leq \frac{\delta^*}{2}, \delta^k \geq \frac{\delta^*}{2}$. Using these results, we can rework the proofs of Theorems 3.3 and 3.6 to get the following corollary.

**Corollary 3.6.1.** *Consider a MPG that satisfies Assumption 3.2 with NPG update using algorithm 7. There exists $K'$, such that for any $K \geq 1$, choosing $\eta = \frac{(1-\gamma)^2}{2\sqrt{n}\phi_{max}}$, we have*

$$\frac{1}{K} \sum_{k=0}^{K-1} NE\text{-}gap(\pi^k) \leq \frac{2M\phi_{max}}{K(1-\gamma)}\left(1 + \frac{4\sqrt{n}\phi_{max}}{c\delta^*(1-\gamma)} + \frac{K'}{2M}\right),$$

*where $M, c$ are defined as in Theorem 3.6.*

## 4   Experimental Results

In previous sections, we established the theoretical convergence of NPG in MPGs. In order to verify the results, we construct two experimental settings for the NPG update and compare our empirical results with existing algorithms. We consider a synthetic potential game scenario with randomly generated rewards and a congestion problem studied in [19] and [8]. We also provide the source code[4] for all experiments.

---

[4] https://github.com/sundave1998/Independent-NPG-MPG

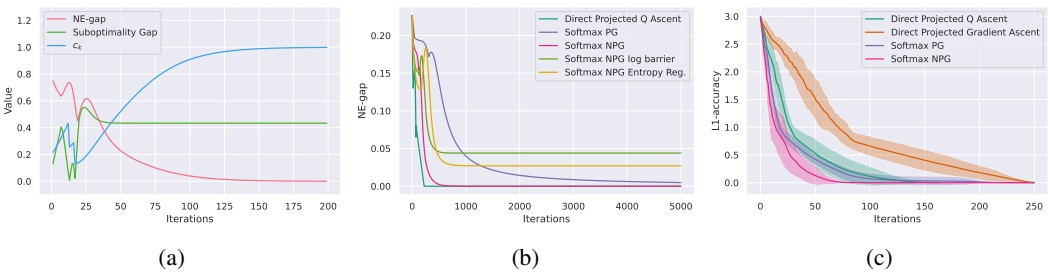

Figure 1: (a) The suboptimality gap; (b) Learning curve in synthetic experiments; (c) Learning curve for congestion game.

### 4.1 Synthetic Potential Games

We first construct a potential game with a fully cooperative objective, where the reward tensor $r(\boldsymbol{a})$ is randomly generated from a uniform distribution. We set the agent number to $n = 3$, each with different action space $|\mathcal{A}_1| = 3, |\mathcal{A}_2| = 4, |\mathcal{A}_3| = 5$. At the start of the experiment, all agents are initialized with uniform policies. We note that the experimental setting in [38] falls under this broad setting, although the experiment therein was a two-player game with carefully designed rewards.

The results are shown in Figure 1b. We compare the performance of independent NPG with other commonly studied policy updates, such as projected Q ascent [8], entropy regularized NPG [38] as well as log-barrier regularized NPG method [38]. We set the learning rate of all algorithms as $\eta = 0.1$. We also fine-tune regularization parameters to find the entropy regularization factor $\tau = 0.05$ and the log-barrier regularization factor $\lambda = 0.005$. As discussed in Section 3.1 (the effect of $c$), we observe in Figure 1b that the regularized algorithms fail to reach an exact Nash policy despite exhibiting good convergence performance at the start. The empirical results here align with our theoretical findings in Section 3.

**Impact of $c$ and $\delta$** We demonstrate the impact of the initialization dependent factor $c^k$ and suboptimality gap $\delta^k$ in MPGs with the same experimental setup. Figure 1a depicts the change in value for $c^k, \delta^k$, and the NE-gap. We can see from the figure that as the algorithm updates over iterations, the value of $c^k$ increases and approaches one, while $\delta^k$ approaches some non-zero constant. Figure 1a shows that although $\delta^k$ could be arbitrarily small in theory if we do not impose technical assumptions, it is not the case in general. Therefore, one should not automatically assume that the suboptimality gap diminishes with respect to iteration, as the limit entirely depends on the problem environment.

The effect of suboptimality gap $\delta^*$ in PG is illustrated in Section E of the appendix under a set of carefully constructed numerical examples. We verify that a larger gap indicates a faster convergence, which corroborates our theory.

### 4.2 Congestion Game

We now consider a class of MDPs where each state defines a congestion game. We borrow the specific settings for this experiment from [19] [8].

For the congestion game experiments, we consider the agent number $n = 8$ with the number of facilities $|\mathcal{A}_i| = 4$, where $\mathcal{A}_i = \{A, B, C, D\}$ as the corresponding individual action spaces. There are two states defined as $\mathcal{S} = \{safe, distancing\}$. In each state, all agents prefer to be taking the same action with as many agents as possible. The reward for an agent selecting action $k$ is defined by predefined weights $w_s^k$ multiplied by the number of other agents taking the same action. Additionally, we set $w_s^A < w_s^B < w_s^C < w_s^D$ and the reward in *distancing* state is reduced by some constant compared to the *safe* state. The state transition depends on the joint actions of all agents. If more than half of all agents take the same action, the system enters a *distancing* state with lower rewards. If the agents are evenly distributed over all actions, the system enters *safe* state with higher rewards.

We use episodic updates with $T = 20$ steps and collect 20 trajectories in each mini-batch and estimate the value function and Q-functions as well as the discounted visitation distribution. We use a discount factor of $\gamma = 0.99$. We adopt the same step size used in [19, 8] and determine optimal step-sizes of

softmax PG and NPG with grid-search. Since regularized methods in Section 4.1 generally do not converge to Nash policies, they are excluded in this experiment. To make the experiment results align with previous works, we provide the $L_1$ distance between the current-iteration policies compared to Nash policies. We plot the mean and variance of $L_1$ distance across multiple runs in Figure 1c. Compared to the direct parameterized algorithms, the two softmax parameterized algorithms exhibit faster convergence, and softmax parameterized NPG has the best performance across all tested algorithms.

## 5   Conclusion and Discussion

In this paper, we studied Markov potential games in the context of multi-agent reinforcement learning. We focused on the independent natural policy gradient algorithm and studied its convergence rate to the Nash equilibrium. The main theorem of the paper shows that the convergence rate of NPG in MPGs is $\mathcal{O}(1/K)$, which improves upon the previous results. Additionally, we provided detailed discussions on the impact of some problem factors (e.g., $c$ and $\delta$) and compared our rate with the best known results with respect to these factors. Two empirical results were presented as a verification of our analysis.

Despite our newly proposed results, there are still many open problems that need to be addressed. One of the limitations of this work is the assumption of Markov potential games, the relaxation of which could extend our analysis to more general stochastic games. As a matter of fact, the gradient-based algorithm studied in this work will fail for a zero-sum game as simple as Tic-Tac-Toe. A similar analysis could also be applied to regularized games and potentially sharper bounds could be obtained. The agents are also assumed to receive gradient information from an oracle in this paper. When such oracle is unavailable, the gradient can be estimated via trajectory samples, which we leave as a future work. Other future directions are the convergence analysis of policy gradient-based algorithms in safe MARL and robust MARL, following the recent exploration of safe single-agent RL [9, 22, 41] and robust single-agent RL [21, 40].

## Acknowledgments and Disclosure of Funding

This material is based upon work partially supported by the US Army Contracting Command under W911NF-22-1-0151 and W911NF2120064, US National Science Foundation under CMMI-2038625, and US Office of Naval Research under N00014-21-1-2385. The views expressed herein and conclusions contained in this document are those of the authors and should not be interpreted as representing the views or official policies, either expressed or implied, of the U.S. NSF, ONR, ARO, or the United States Government. The U.S. Government is authorized to reproduce and distribute reprints for Government purposes notwithstanding any copyright notation herein.

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

# A    Proof for Potential Games in Section 3.1

We first find the upper bound on the difference in marginalized rewards under two policies. The following lemma establishes a "smooth" relationship between the marginalized rewards and their respective policies. For ease of notation, we denote $r(\pi) := \mathbb{E}_{\boldsymbol{a}\sim\pi}[r(\boldsymbol{a})]$ and $\phi(\pi) := \mathbb{E}_{\boldsymbol{a}\sim\pi}[\phi(\boldsymbol{a})]$, where these function values are taken expectation over policy $\pi$.

**Lemma A.1.** *For any two sets of policies $\pi, \pi' \in \Delta(\mathcal{A}_1)\times...\times\Delta(\mathcal{A}_n)$, the difference in marginalized reward of any agent $i$ is bounded by the total variation distance of policies,*

$$\|\bar{r}_i^\pi - \bar{r}_i^{\pi'}\|_\infty \leq 2TV(\pi_{-i}, \pi'_{-i}).$$

*Proof.* Given any $\pi, \pi' \in \Delta(\mathcal{A})$, we have

$$|\bar{r}_i^\pi(a_i) - \bar{r}_i^{\pi'}(a_i)| = |\mathbb{E}_{a_{-i}\sim\pi_{-i}}[r_i(a_i, a_{-i})] - \mathbb{E}_{a_{-i}\sim\pi'_{-i}}[r_i(a_i, a_{-i})]|$$
$$\leq \|r_i\|_\infty\|\pi_{-i} - \pi'_{-i}\|_1 \leq 2TV(\pi_{-i}, \pi'_{-i}).$$

$\square$

Next, we provide the proof of Lemma 3.1, which demonstrates the lower bound of the potential function difference.

**Lemma A.2** (Restatement of Lemma 3.1). *Given policy $\pi^k$ and marginalized reward $\bar{r}_i^k(a_i)$, for $\pi^{k+1}$ generated using the NPG update in (4), we have the following inequality for any $\eta < \frac{1}{\sqrt{n}}$,*

$$\phi(\pi^{k+1}) - \phi(\pi^k) \geq (1 - \sqrt{n}\eta)\sum_{i=1}^n \langle \pi_i^{k+1} - \pi_i^k, \bar{r}_i^k \rangle.$$

*Proof of Lemma 3.1.* We introduce

$$\tilde{\pi}_{-i}^k(a_{-i}) := \prod_{j<i}\pi_j^k(a_j)\prod_{\ell>i}\pi_\ell^{k+1}(a_\ell)$$

to denote the mixed strategy where any agent with index $j < i$ follows $\pi_j^k$ and any agent with index $\ell > i$ follows $\pi_\ell^{k+1}$ instead. Let $\tilde{r}_i^k$ be the associated marginalized reward function, i.e.,

$$\tilde{r}_i^k(a_i) = \mathbb{E}_{a_{-i}\sim\tilde{\pi}_{-i}^k}[r_i(\boldsymbol{a})] = \sum_{a_{-i}\in\mathcal{A}_{-i}} r_i(a_i, a_{-i})\prod_{j<i}\pi_j^k(a_j)\prod_{\ell>i}\pi_\ell^{k+1}(a_\ell).$$

It follows that

$$\phi\left(\pi_i^k, \tilde{\pi}_{-i}^k\right) = \phi\left(\pi_1^k, \cdots, \pi_i^k, \pi_{i+1}^{k+1}, \cdots, \pi_n^{k+1}\right) = \phi\left(\pi_{i+1}^{k+1}, \tilde{\pi}_{-(i+1)}^k\right).$$

We now decompose $\phi^{k+1} - \phi^k$ as follows:

$$\phi(\pi^{k+1}) - \phi(\pi^k) = \phi\left(\pi_1^{k+1}, \tilde{\pi}_{-1}^k\right) - \phi\left(\pi_n^k, \tilde{\pi}_{-n}^k\right)$$
$$= \sum_{i=1}^n \phi\left(\pi_i^{k+1}, \tilde{\pi}_{-i}^k\right) - \phi\left(\pi_i^k, \tilde{\pi}_{-i}^k\right)$$
$$= \sum_{i=1}^n r_i\left(\pi_i^{k+1}, \tilde{\pi}_{-i}^k\right) - r_i\left(\pi_i^k, \tilde{\pi}_{-i}^k\right)$$
$$= \sum_{i=1}^n \langle \tilde{r}_i^k - \bar{r}_i^k, \pi_i^{k+1} - \pi_i^k \rangle + \sum_{i=1}^n \langle \bar{r}_i^k, \pi_i^{k+1} - \pi_i^k \rangle.$$

The third equality follows from the definition (3).

Since

$$\sum_{i=1}^n |\langle \tilde{r}_i^k - \bar{r}_i^k, \pi_i^{k+1} - \pi_i^k \rangle| \leq 2\sum_{i=1}^n TV(\tilde{\pi}_{-i}^k, \pi_{-i}^k)\|\pi_i^{k+1} - \pi_i^k\|_1$$

$$\leq \frac{2}{\sqrt{n}} \sum_{i=1}^{n} TV(\tilde{\pi}_{-i}^k, \pi_{-i}^k)^2 + \frac{\sqrt{n}}{2} \sum_{i=1}^{n} \|\pi_i^{k+1} - \pi_i^k\|_1^2$$

$$\leq \frac{1}{\sqrt{n}} \sum_{i=1}^{n} KL(\pi_{-i}^k || \tilde{\pi}_{-i}^k) + \sqrt{n} \sum_{i=1}^{n} KL(\pi_i^{k+1} || \pi_i^k)$$

$$\leq \sqrt{n}[KL(\pi^k || \pi^{k+1}) + KL(\pi^{k+1} || \pi^k)]$$

$$= \sqrt{n} \sum_{i=1}^{n} \langle \pi_i^{k+1} - \pi_i^k, \log \pi_i^{k+1} - \log \pi_i^k \rangle$$

$$= \eta \sqrt{n} \sum_{i=1}^{n} \langle \pi_i^{k+1} - \pi_i^k, \bar{r}_i^k \rangle,$$

we have

$$\phi(\pi^{k+1}) - \phi(\pi^k) \geq (1 - \sqrt{n}\eta) \sum_{i=1}^{n} \langle \pi_i^{k+1} - \pi_i^k, \bar{r}_i^k \rangle.$$

$\square$

Additionally, we provide the proof of Lemma 3.2, which builds the relationship between potential function difference and NE-gap.

**Lemma A.3** (Restatement of Lemma 3.2). *For function $f^k(\alpha)$ defined in (5) and any $\alpha > 0$, we have the following inequality.*

$$f^k(\alpha) \geq f^k(\infty) \left[ 1 - \frac{1}{c(\exp(\alpha \delta_K) - 1) + 1} \right].$$

*Proof.* Recall that we define $a_{i_p}^k \in \arg\max_{a_j \in \mathcal{A}_i} \bar{r}_i^k(a_j) =: \mathcal{A}_{i_p}^k$ and $a_{i_q}^k \in \arg\max_{a_j \in \mathcal{A}_i \backslash \mathcal{A}_{i_p}^k} \bar{r}_i^k(a_j)$, where $\mathcal{A}_{i_p}^k$ denotes the set of the best possible actions for agent $i$. For any $i \in [n]$, we have

$$\langle \pi_{i,\alpha} - \pi_i^k, \bar{r}_i^k \rangle$$

$$= \frac{\sum_{a_i} \bar{r}_i^k(a_i) \pi_i^k(a_i) \exp\{\bar{r}_i^k(a_i)\alpha\}}{\sum_{a_i} \pi_i^k(a_i) \exp\{\bar{r}_i^k(a_i)\alpha\}} - \sum_{a_i} \pi_i^k(a_i) \bar{r}_i^k(a_i)$$

$$\geq \frac{\sum_{a_j \in \mathcal{A}_{i_p}^k} \bar{r}_i^k(a_j) \pi_i^k(a_j) \exp\{\bar{r}_i^k(a_j)\alpha\} + \sum_{a_j \in \mathcal{A}_i \backslash \mathcal{A}_{i_p}^k} \bar{r}_i^k(a_j) \pi_i^k(a_j) \exp\{\bar{r}_i^k(a_{i_q}^k)\alpha\}}{\sum_{a_j \in \mathcal{A}_{i_p}^k} \pi_i^k(a_j) \exp\{\bar{r}_i^k(a_j)\alpha\} + \sum_{a_j \in \mathcal{A}_i \backslash \mathcal{A}_{i_p}^k} \pi_i^k(a_j) \exp\{\bar{r}_i^k(a_{i_q}^k)\alpha\}} - \sum_{a_i} \pi_i^k(a_i) \bar{r}_i^k(a_i)$$

$$= \frac{\bar{r}_i^k(a_{i_p}^k) \left(\sum_{a_j \in \mathcal{A}_{i_p}^k} \pi_i^k(a_j)\right) \exp\{(\bar{r}_i^k(a_{i_p}^k) - \bar{r}_i^k(a_{i_q}^k))\alpha\} + \sum_{a_j \in \mathcal{A}_i \backslash \mathcal{A}_{i_p}^k} \bar{r}_i^k(a_j) \pi_i^k(a_j)}{\left(\sum_{a_j \in \mathcal{A}_{i_p}^k} \pi_i^k(a_j)\right) \exp\{(\bar{r}_i^k(a_{i_p}^k) - \bar{r}_i^k(a_{i_q}^k))\alpha\} + \sum_{a_j \in \mathcal{A}_i \backslash \mathcal{A}_{i_p}^k} \pi_i^k(a_j)} - \sum_{a_i} \pi_i^k(a_i) \bar{r}_i^k(a_i)$$

$$= \frac{\bar{r}_i^k(a_{i_p}^k) \left(\sum_{a_j \in \mathcal{A}_{i_p}^k} \pi_i^k(a_j)\right) \exp\{(\bar{r}_i^k(a_{i_p}^k) - \bar{r}_i^k(a_{i_q}^k))\alpha\} + \sum_{a_j \in \mathcal{A}_i \backslash \mathcal{A}_{i_p}^k} \bar{r}_i^k(a_{i_p}^k) \pi_i^k(a_j)}{\left(\sum_{a_j \in \mathcal{A}_{i_p}^k} \pi_i^k(a_j)\right) \exp\{(\bar{r}_i^k(a_{i_p}^k) - \bar{r}_i^k(a_{i_q}^k))\alpha\} + \sum_{a_j \in \mathcal{A}_i \backslash \mathcal{A}_{i_p}^k} \pi_i^k(a_j)}$$

$$- \frac{\sum_{a_j \in \mathcal{A}_i \backslash \mathcal{A}_{i_p}^k} (\bar{r}_i^k(a_{i_p}^k) - \bar{r}_i^k(a_j)) \pi_i^k(a_j)}{\left(\sum_{a_j \in \mathcal{A}_{i_p}^k} \pi_i^k(a_j)\right) \exp\{(\bar{r}_i^k(a_{i_p}^k) - \bar{r}_i^k(a_{i_q}^k))\alpha\} + \sum_{a_j \in \mathcal{A}_i \backslash \mathcal{A}_{i_p}^k} \pi_i^k(a_j)} - \sum_{a_i} \pi_i^k(a_i) \bar{r}_i^k(a_i)$$

$$= \bar{r}_i^k(a_{i_p}^k) - \sum_{a_i} \pi_i^k(a_i) \bar{r}_i^k(a_i) - \frac{\sum_{a_i} (\bar{r}_i^k(a_{i_p}^k) - \bar{r}_i^k(a_i)) \pi_i^k(a_i)}{\left(\sum_{a_j \in \mathcal{A}_{i_p}^k} \pi_i^k(a_j)\right) \left(\exp\{(\bar{r}_i^k(a_{i_p}^k) - \bar{r}_i^k(a_{i_q}^k))\alpha\} - 1\right) + 1}$$

$$= \left(\sum_{a_i} (\bar{r}_i^k(a_{i_p}^k) - \bar{r}_i^k(a_i)) \pi_i^k(a_i)\right) \left[ 1 - \frac{1}{\left(\sum_{a_j \in \mathcal{A}_{i_p}^k} \pi_i^k(a_j)\right) \left(\exp\{(\bar{r}_i^k(a_{i_p}^k) - \bar{r}_i^k(a_{i_q}^k))\alpha\} - 1\right) + 1} \right]$$

$$\geq \langle \pi_i^{*k} - \pi_i^k, \bar{r}_i^k \rangle \left[ 1 - \frac{1}{c(\exp\{\delta^k \alpha\} - 1) + 1} \right],$$

where the first inequality holds due to Lemma D.1. By taking sum over all agents $i \in [n]$, we get

$$f^k(\alpha) = \sum_{i \in [n]} \langle \pi_{i,\alpha} - \pi_i^k, \bar{r}_i^k \rangle$$

$$\geq \sum_{i \in [n]} \langle \pi_i^{*k} - \pi_i^k, \bar{r}_i^k \rangle \left[ 1 - \frac{1}{c(\exp\{\delta^k \alpha\} - 1) + 1} \right]$$

$$\geq f^k(\infty) \left[ 1 - \frac{1}{c(\exp\{\delta_K \alpha\} - 1) + 1} \right].$$

$\square$

We next provide the following lemma on the structure of potential games. This lemma is not directly used in the analysis of this paper but instead provides a discussion on the function class of potential games, which could be of separate interest.

**Lemma A.4** (Structure of potential game reward function class). *For any agent $i$, the set of reward functions for potential games, $\mathcal{R}_i$, has the following structure within the ambient space of $\mathbb{R}^{\prod_j |\mathcal{A}_j|}$.*

$$\mathcal{R}_i = \left\{ \phi(a_i, a_{-i}) + g_i(a_{-i}) : \forall g_i : \prod_{j \neq i} \mathcal{A}_j \to \mathbb{R} \right\} \subset \mathbb{R}^{\prod_j |\mathcal{A}_j|}. \tag{8}$$

*Proof.* The reward vector $r_i \in \mathbb{R}^{\prod_{i=1}^n |\mathcal{A}_i|}$ can be viewed as an $\prod_{i=1}^n |\mathcal{A}_i|$-dimensional linear vector. There are in total $(|\mathcal{A}_i| - 1) \prod_{j \neq i} |\mathcal{A}_j| = \prod_{i=1}^n |\mathcal{A}_i| - \prod_{j \neq i} |\mathcal{A}_j|$ independent constraints w.r.t $r_i$ and potential function $\phi$. Since $\mathcal{R}_i$ is a linear space with dimension $\prod_{j \neq i} |\mathcal{A}_j|$, $\mathcal{R}_i$ is the space of valid reward functions for potential games. $\square$

# B  Proof for Markov Potential Games in Section 3.2

We first introduce the following lemma to fill the gap between general MPGs and fully-cooperative MPGs. Define $h_i(s, \boldsymbol{a}) := r_i(s, \boldsymbol{a}) - \phi(s, \boldsymbol{a})$, which implies $V_i^\pi(s) = \Phi^\pi(s) + V_{h_i}^\pi(s)$.

**Lemma B.1** (Restatement of Lemma 3.4). *Define $\bar{A}_{h_i}^\pi(s, a_i)$ with respect to $h_i$ similar to (2). We then have*

$$\sum_{a_i} (\pi_i'(a_i|s) - \pi_i(a_i|s)) \bar{A}_{h_i}^\pi(s, a_i) = 0, \quad \forall s \in \mathcal{S}, i \in [n], \pi_i', \pi_i \in \Delta(\mathcal{A}_i).$$

*Proof of Lemma 3.4.* Due to the definition of Markov potential game, we have

$$V_i^{\pi_i', \pi_{-i}}(\mu) - V_i^{\pi_i, \pi_{-i}}(\mu) = \Phi^{\pi_i', \pi_{-i}}(\mu) - \Phi^{\pi_i, \pi_{-i}}(\mu),$$

for any $\mu$, $\pi_i'$ and $\pi_i$. It then follows that

$$0 = V_{h_i}^{\pi_i', \pi_{-i}}(\mu) - V_{h_i}^{\pi_i, \pi_{-i}}(\mu) = \frac{1}{1 - \gamma} \sum_s d_\mu^{\pi_i', \pi_{-i}}(s) \sum_{a_i} (\pi_i'(a_i|s) - \pi_i(a_i|s)) \bar{A}_{h_i}^\pi(s, a_i).$$

Denote $[\tilde{P}]_{s,s'} = \mathbb{E}_{a \sim \pi_i' \times \pi_{-i}}[P(s'|s, a)]$ as the transition matrix generated by policy $\pi_i' \times \pi_{-i}$. Then

$$\frac{1}{1 - \gamma} d_\mu^{\pi_i', \pi_{-i}} = \mu(I + \gamma \tilde{P} + \gamma^2 \tilde{P}^2 + \cdots) = \mu(I - \gamma \tilde{P})^{-1},$$

where $d_\mu^{\pi_i', \pi_{-i}}$ and $\mu$ are row vectors and $(I - \gamma \tilde{P})^{-1}$ is clearly full rank. Since the above relationship holds for any $\mu \in \Delta_\mathcal{S}$, we get $0 = \sum_{a_i} (\pi_i'(a_i|s) - \pi_i(a_i|s)) \bar{A}_{h_i}^\pi(s, a_i), \quad \forall s \in \mathcal{S}.$ $\square$

Then, we analyze the lower bound of the potential value function difference.

**Lemma B.2** (Restatement of Lemma 3.5). *Given policy $\pi^k$ and marginalized advantage function $\bar{A}_i^{\pi^k}(s, a_i)$, for $\pi^{k+1}$ generated using NPG update in (7), we have the following inequality,*

$$\Phi^{\pi^{k+1}}(\rho) - \Phi^{\pi^k}(\rho) \geq \left(\frac{1}{1-\gamma} - \frac{\sqrt{n}\phi_{max}\eta}{(1-\gamma)^3}\right) \sum_s d_\rho^{\pi^{k+1}}(s) \sum_{i=1}^n \langle \pi_i^{k+1}(\cdot|s), \bar{A}_i^{\pi^k}(s, \cdot)\rangle.$$

*Proof of Lemma 3.5.* We introduce

$$\tilde{\pi}_{-i}^k(a_{-i}|s) := \prod_{j<i} \pi_j^{k+1}(a_j|s) \prod_{\ell>i} \pi_\ell^k(a_\ell|s)$$

to denote the mixed strategy where each agent with index $j < i$ follows $\pi_j^{k+1}$ and each agent with index $\ell > i$ follows $\pi_\ell^k$ instead. Let $\bar{A}_{i,\phi}^{\tilde{\pi}^k}$ be the associated marginalized advantage value based on potential function, i.e.,

$$\bar{A}_{i,\phi}^{\tilde{\pi}^k}(s, a_i) = \sum_{a_{-i}\in\mathcal{A}_{-i}} A_\phi^{\pi^k}(s, a_i, a_{-i}) \prod_{j<i} \pi_j^{k+1}(a_j|s) \prod_{\ell>i} \pi_\ell^k(a_\ell|s).$$

We now decompose $\Phi^{\pi^{k+1}} - \Phi^{\pi^k}$ as follows:

$$\Phi^{\pi^{k+1}}(\rho) - \Phi^{\pi^k}(\rho)$$

$$= \frac{1}{1-\gamma} \sum_s d_\rho^{\pi^{k+1}}(s) \sum_a (\pi^{k+1}(a|s) - \pi^k(a|s))A_\phi^{\pi^k}(s, a)$$

$$= \frac{1}{1-\gamma} \sum_s d_\rho^{\pi^{k+1}}(s) \sum_a \sum_{i=1}^n \left(\prod_{j=1}^i \pi_j^{k+1}(a_j|s) \prod_{\ell=i+1}^n \pi_\ell^k(a_\ell|s) - \prod_{j=1}^{i-1} \pi_j^{k+1}(a_j|s)\prod_{\ell=i}^n \pi_\ell^k(a_\ell|s)\right) A_\phi^{\pi^k}(s, a)$$

$$= \frac{1}{1-\gamma} \sum_s d_\rho^{\pi^{k+1}}(s) \sum_{i=1}^n \sum_{a_i}(\pi_i^{k+1}(a_i|s) - \pi_i^k(a_i|s))\bar{A}_{i,\phi}^{\tilde{\pi}^k}(s, a_i)$$

$$= \frac{1}{1-\gamma} \sum_s d_\rho^{\pi^{k+1}}(s) \sum_{i=1}^n \sum_{a_i}(\pi_i^{k+1}(a_i|s) - \pi_i^k(a_i|s))\bar{A}_i^{\pi^k}(s, a_i)$$

$$\quad - \frac{1}{1-\gamma} \sum_s d_\rho^{\pi^{k+1}}(s) \sum_{i=1}^n \sum_{a_i}(\pi_i^{k+1}(a_i|s) - \pi_i^k(a_i|s))\bar{A}_{h_i}^{\pi^k}(s, a_i)$$

$$\quad + \frac{1}{1-\gamma} \sum_s d_\rho^{\pi^{k+1}}(s) \sum_{i=1}^n \sum_{a_i}(\pi_i^{k+1}(a_i|s) - \pi_i^k(a_i|s))(\bar{A}_{i,\phi}^{\tilde{\pi}^k}(s, a_i) - \bar{A}_{i,\phi}^{\pi^k}(s, a_i)).$$

Since

$$|\bar{A}_{i,\phi}^{\tilde{\pi}^k}(s, a_i) - \bar{A}_{i,\phi}^{\pi^k}(s, a_i)|$$

$$= \left|\sum_{a_{-i}} \left(\prod_{j=1}^{i-1} \pi_j^{k+1}(a_j|s) - \prod_{j=1}^{i-1}\pi_j^k(a_j|s)\right) \prod_{\ell=i+1}^n \pi_\ell^k(a_\ell|s) A_\phi^{\pi^k}(s, \boldsymbol{a})\right|$$

$$\leq \frac{\phi_{max}}{1-\gamma}\|\tilde{\pi}_{-i}^k(\cdot|s) - \pi_{-i}^k(\cdot|s)\|_1,$$

using Lemma 3.4, we have

$$\Phi^{\pi^{k+1}}(\rho) - \Phi^{\pi^k}(\rho) \geq \frac{1}{1-\gamma} \sum_s d_\rho^{\pi^{k+1}}(s) \sum_{i=1}^n \sum_{a_i} \pi_i^{k+1}(a_i|s)\bar{A}_i^{\pi^k}(s, a_i)$$

$$\quad - \frac{\phi_{max}}{(1-\gamma)^2} \sum_s d_\rho^{\pi^{k+1}}(s) \sum_{i=1}^n \|\pi_i^{k+1}(\cdot|s) - \pi_i^k(\cdot|s)\|_1\|\tilde{\pi}_{-i}^k(\cdot|s) - \pi_{-i}^k(\cdot|s)\|_1.$$

Since

$$\sum_{i=1}^{n} \|\pi_i^{k+1}(\cdot|s) - \pi_i^k(\cdot|s)\|_1 \|\tilde{\pi}_{-i}^k(\cdot|s) - \pi_{-i}^k(\cdot|s)\|_1$$

$$\leq \frac{\sqrt{n}}{2} \sum_{i=1}^{n} \|\pi_i^{k+1}(\cdot|s) - \pi_i^k(\cdot|s)\|_1^2 + \frac{1}{2\sqrt{n}} \sum_{i=1}^{n} \|\tilde{\pi}_{-i}^k(\cdot|s) - \pi_{-i}^k(\cdot|s)\|_1^2$$

$$\leq \sqrt{n} \sum_{i=1}^{n} \left( KL(\pi_i^{k+1}(\cdot|s)\|\pi_i^k(\cdot|s)) + KL(\pi_i^k(\cdot|s)\|\pi_i^{k+1}(\cdot|s)) \right)$$

$$= \sqrt{n} \sum_{i=1}^{n} \langle \pi_i^{k+1}(\cdot|s) - \pi_i^k(\cdot|s), \log \pi_i^{k+1}(\cdot|s) - \log \pi_i^k(\cdot|s) \rangle$$

$$= \frac{\sqrt{n}\eta}{1-\gamma} \sum_{i=1}^{n} \langle \pi_i^{k+1}(\cdot|s) - \pi_i^k(\cdot|s), \bar{A}_i^{\pi^k}(s, \cdot) \rangle = \frac{\sqrt{n}\eta}{1-\gamma} \sum_{i=1}^{n} \langle \pi_i^{k+1}(\cdot|s), \bar{A}_i^{\pi^k}(s, \cdot) \rangle,$$

we have

$$\Phi^{\pi^{k+1}}(\rho) - \Phi^{\pi^k}(\rho) \geq \left( \frac{1}{1-\gamma} - \frac{\sqrt{n}\phi_{max}\eta}{(1-\gamma)^3} \right) \sum_s d_\rho^{\pi^{k+1}}(s) \sum_{i=1}^{n} \langle \pi_i^{k+1}(\cdot|s), \bar{A}_i^{\pi^k}(s, \cdot) \rangle.$$

$\square$

Next, we introduce the following lemma, which is the MPG version of Lemma 3.2.

**Lemma B.3.** *At iteration $k$, define*

$$f^k(\alpha) = \sum_{i \in [n]} \sum_{a_i} \pi_{i,\alpha}(a_i|s) \bar{A}_i^{\pi^k}(s, a_i), \text{ where } \pi_{i,\alpha}(\cdot|s) \propto \pi_i^k(\cdot|s) \exp\left\{ \frac{\alpha \bar{A}_i^{\pi^k}(s, \cdot)}{1-\gamma} \right\}.$$

*Then, for any $\alpha > 0$,*

$$f^k(\alpha) \geq f^k(\infty) \left[ 1 - \frac{1}{c(\exp\left(\frac{\delta^k \alpha}{1-\gamma}\right) - 1) + 1} \right],$$

*with $c$ and $\delta^k$ defined in Theorem 3.6.*

*Proof.* For any $i \in [n]$, we have

$$f^k(\alpha) = \sum_{i \in [n]} \frac{\sum_{a_i} \bar{A}_i^{\pi^k}(s, a_i)\pi_i^k(a_i|s) \exp\left\{ \frac{\alpha \bar{A}_i^{\pi^k}(s,a_i)}{1-\gamma} \right\}}{\sum_{a_i} \pi_i^k(a_i|s) \exp\left\{ \frac{\alpha \bar{A}_i^{\pi^k}(s,a_i)}{1-\gamma} \right\}}$$

$$\geq \sum_{i \in [n]} \frac{\bar{A}_i^{\pi^k}(s, a_{i_p}^k) \left( \sum_{a_j \in \mathcal{A}_{i_p}^k} \pi_i^k(a_j|s) \right) \exp\left\{ \frac{\alpha \bar{A}_i^{\pi^k}(s,a_{i_p}^k)}{1-\gamma} \right\} + \sum_{a_j \in \mathcal{A}_i \setminus \mathcal{A}_{i_p}^k} \bar{A}_i^{\pi^k}(s, a_j)\pi_i^k(a_j|s) \exp\left\{ \frac{\alpha \bar{A}_i^{\pi^k}(s,a_{i_q}^k)}{1-\gamma} \right\}}{\left( \sum_{a_j \in \mathcal{A}_{i_p}^k} \pi_i^k(a_j|s) \right) \exp\left\{ \frac{\alpha \bar{A}_i^{\pi^k}(s,a_{i_p}^k)}{1-\gamma} \right\} + \sum_{a_j \in \mathcal{A}_i \setminus \mathcal{A}_{i_p}^k} \pi_i^k(a_j|s) \exp\left\{ \frac{\alpha \bar{A}_i^{\pi^k}(s,a_{i_q}^k)}{1-\gamma} \right\}}$$

$$= \sum_{i \in [n]} \frac{\bar{A}_i^{\pi^k}(s, a_{i_p}^k) \left( \sum_{a_j \in \mathcal{A}_{i_p}^k} \pi_i^k(a_j|s) \right) \exp\left\{ \frac{\alpha(\bar{A}_i^{\pi^k}(s,a_{i_p}^k) - \bar{A}_i^{\pi^k}(s,a_{i_q}^k))}{1-\gamma} \right\} + \sum_{a_j \in \mathcal{A}_i \setminus \mathcal{A}_{i_p}^k} \bar{A}_i^{\pi^k}(s, a_j)\pi_i^k(a_j|s)}{\left( \sum_{a_j \in \mathcal{A}_{i_p}^k} \pi_i^k(a_j|s) \right) \exp\left\{ \frac{\alpha(\bar{A}_i^{\pi^k}(s,a_{i_p}^k) - \bar{A}_i^{\pi^k}(s,a_{i_q}^k))}{1-\gamma} \right\} + \sum_{a_j \in \mathcal{A}_i \setminus \mathcal{A}_{i_p}^k} \pi_i^k(a_j|s)}$$

$$= \sum_{i \in [n]} \left[ \bar{A}_i^{\pi^k}(s, a_{i_p}^k) - \frac{\sum_{a_j \in \mathcal{A}_i \setminus \mathcal{A}_{i_p}^k} (\bar{A}_i^{\pi^k}(s, a_{i_p}^k) - \bar{A}_i^{\pi^k}(s, a_j))\pi_i^k(a_j|s)}{\left( \sum_{a_j \in \mathcal{A}_{i_p}^k} \pi_i^k(a_j|s) \right) \exp\left\{ \frac{\alpha(\bar{A}_i^{\pi^k}(s,a_{i_p}^k) - \bar{A}_i^{\pi^k}(s,a_{i_q}^k))}{1-\gamma} \right\} + \sum_{a_j \in \mathcal{A}_i \setminus \mathcal{A}_{i_p}^k} \pi_i^k(a_j|s)} \right]$$

$$\geq \sum_{i\in[n]} \bar{A}_i^{\pi^k}(s, a_{i_p}^k)\left[1 - \frac{1}{c(\exp\left(\frac{\delta^k\alpha}{1-\gamma}\right)-1)+1}\right]$$

$$\geq f^k(\infty)\left[1 - \frac{1}{c(\exp\left(\frac{\delta^k\alpha}{1-\gamma}\right)-1)+1}\right],$$

where the first inequality holds due to Lemma D.1. $\qquad\square$

Combining the above lemmas, we show the proof of Theorem 3.6.

*Proof of Theorem 3.6.* Choose $\alpha = \eta = \frac{(1-\gamma)^2}{2\sqrt{n}\phi_{max}}$ in Lemma B.3,

$$\text{NE-gap}(\pi^k) \leq \frac{1}{1-\gamma}\sum_{i,s} d_\rho^{\pi_i^{*k}, \pi_{-i}^k}(s)\max_{a_i}\bar{A}_i^{\pi^k}(s, a_i)$$

$$\leq \frac{1}{1-\gamma}\sum_{i,s}\|\frac{d_\rho^{\pi_i^{*k}, \pi_{-i}^k}}{d_\rho^{\pi^{k+1}}}\|_\infty d_\rho^{\pi^{k+1}}(s)\max_{a_i}\bar{A}_i^{\pi^k}(s, a_i)$$

$$\leq \frac{1}{1-\gamma}M\sum_s d_\rho^{\pi^{k+1}}(s)\sum_i\max_{a_i}\bar{A}_i^{\pi^k}(s, a_i)$$

$$\leq \frac{1}{1-\gamma}M\sum_s d_\rho^{\pi^{k+1}}(s)\lim_{\alpha\to\infty}f^k(\alpha)$$

$$\leq \frac{1}{1-\gamma}M\sum_s d_\rho^{\pi^{k+1}}(s)\frac{1}{1-\frac{1}{c(\exp\left(\frac{\delta^k\eta}{1-\gamma}\right)-1)+1}}f^k(\eta)$$

$$\leq \frac{1}{2(1-\gamma)}\sum_s d_\rho^{\pi^{k+1}}(s)\sum_{i=1}^n\sum_{a_i}\pi_i^{k+1}(a_i|s)\bar{A}_i^{\pi^k}(s, a_i)2M(1+\frac{1-\gamma}{c\delta^k\eta})$$

$$\leq (\Phi^{\pi^{k+1}}(\rho) - \Phi^{\pi^k}(\rho))2M(1+\frac{2\sqrt{n}\phi_{max}}{c\delta^k(1-\gamma)})$$

$$\leq (\Phi^{\pi^{k+1}}(\rho) - \Phi^{\pi^k}(\rho))2M(1+\frac{2\sqrt{n}\phi_{max}}{c\delta_K(1-\gamma)}).$$

Then

$$\frac{1}{K}\sum_{k=0}^{K-1}\text{NE-gap}(\pi^k) \leq \frac{2M(\Phi^{\pi^K}(\rho)-\Phi^{\pi^0}(\rho))}{K}(1+\frac{2\sqrt{n}\phi_{max}}{c\delta_K(1-\gamma)}) \leq \frac{2M\phi_{max}}{K(1-\gamma)}(1+\frac{2\sqrt{n}\phi_{max}}{c\delta_K(1-\gamma)}).$$

$\qquad\square$

Finally, we show a corollary of Theorem 3.6.

**Corollary B.3.1** (Restatement of Corollary 3.6.1). *Consider a MPG that satisfies Assumption 3.2 with NPG update using algorithm 7. There exists $K'$, such that for any $K \geq 1$, choosing $\eta = \frac{(1-\gamma)^2}{2\sqrt{n}\phi_{max}}$, we have*

$$\frac{1}{K}\sum_{k=0}^{K-1} NE\text{-}gap(\pi^k) \leq \frac{2M\phi_{max}}{K(1-\gamma)}(1+\frac{4\sqrt{n}\phi_{max}}{c\delta^*(1-\gamma)}+\frac{K'}{2M}).$$

*where $M, c$ are defined as in Theorem 3.6.*

*Proof of Corollary 3.6.1.* Following from the main paper, we know that there exists $K'$ such that for all $k \geq K'$, $\delta^k \geq \frac{\delta^*}{2}$.

From the proof of Theorem 3.6, we know that for MPGs, we get the following relationship:

$$\text{NE-gap}(\pi^k) \leq (\Phi^{\pi^{k+1}}(\rho) - \Phi^{\pi^k}(\rho))2M(1 + \frac{2\sqrt{n}\phi_{max}}{c\delta^k(1-\gamma)}).$$

Therefore, for all $k \geq K'$,

$$\text{NE-gap}(\pi^k) \leq (\Phi^{\pi^{k+1}}(\rho) - \Phi^{\pi^k}(\rho))2M(1 + \frac{4\sqrt{n}\phi_{max}}{c\delta^*(1-\gamma)}).$$

Then, taking summation from $K'$ to $K-1$, we get

$$\sum_{k=K'}^{K-1} \text{NE-gap}(\pi^k) \leq \frac{2M\phi_{max}}{1-\gamma}(1 + \frac{4\sqrt{n}\phi_{max}}{c\delta^*(1-\gamma)}),$$

Additionally, we know that $\text{NE-gap}(\pi^k) \leq \frac{\phi_{max}}{1-\gamma}$ for all $k$ by the definition of MPG. Combining the equations above, we have

$$
\begin{aligned}
\frac{1}{K}\sum_{k=0}^{K-1} \text{NE-gap}(\pi^k) &= \frac{1}{K}\left(\sum_{k=0}^{K'-1}\text{NE-gap}(\pi^k) + \sum_{k=K'}^{K-1}\text{NE-gap}(\pi^k)\right) \\
&\leq \frac{1}{K}\left(\frac{2M\phi_{max}}{(1-\gamma)}(1 + \frac{4\sqrt{n}\phi_{max}}{c\delta^*(1-\gamma)}) + \frac{\phi_{max}K'}{1-\gamma}\right) \\
&= \frac{2M\phi_{max}}{K(1-\gamma)}(1 + \frac{4\sqrt{n}\phi_{max}}{c\delta^*(1-\gamma)} + \frac{K'}{2M}).
\end{aligned}
$$

$\square$

## C  Extension to General MPG Formulation

In this section, we extend our results to a more general form of potential function.

*Note that the detailed analysis is omitted for brevity. Proofs of lemmas and theorem in this section either exist in previous works or closely follow our analysis in Section B.*

**Definition C.1** (Alternate definition of MPG [8]). *For a MPG, at any state $s$, there exists a global function $\Phi^\pi(s) : \Pi \times \mathcal{S} \to \mathbb{R}$ such that*

$$V_i^{\pi_i, \pi_{-i}}(s) - V_i^{\pi_i', \pi_{-i}}(s) = \Phi^{\pi_i, \pi_{-i}}(s) - \Phi^{\pi_i', \pi_{-i}}(s) \tag{9}$$

*is true for any policies $\pi_i, \pi_i' \in \Pi_i$.*

This definition of MPG differs from Definition 2.1 and does not enforce the existence of additional structure in function $\phi$. However, whether the two definitions are equivalent, similar to Lemma A.4, is still an open question to the best of the authors' knowledge.

We present the following Lemmas from [8], using which we provide a similar result to Lemma 3.5.

**Lemma C.1** ([8]). *For any function $\Psi^\pi : \Pi \to \mathbb{R}$, and any two policies $\pi, \pi' \in \Pi$,*

$$
\Psi^{\pi'} - \Psi^\pi = \sum_{i=1}^{N}(\Psi^{\pi_i', \pi_{-i}} - \Psi^\pi)
$$
$$
+ \sum_{i=1}^{N}\sum_{j=i+1}^{N}\left(\Psi^{\pi_{<i,i\ j},\pi_{>j}',\pi_i',\pi_j'} - \Psi^{\pi_{<i,i\ j},\pi_{>j}',\pi_i,\pi_j'} - \Psi^{\pi_{<i,i\ j},\pi_{>j}',\pi_i',\pi_j} + \Psi^{\pi_{<i,i\ j},\pi_{>j}',\pi_i,\pi_j}\right)
$$

**Lemma C.2** ([8]). *Consider a two-player game (group of players) with a common-payoff Markov game with state space $\mathcal{S}$ and action sets $\mathcal{A}_1, \mathcal{A}_2$, with reward function $r$, transition function $P$ and policy sets $\Pi_1, \Pi_2$ be defined with respect to general MDPs. Then for any $\pi_1, \pi_1' \in \Pi_1, \pi_2, \pi_2' \in \Pi_2$, the for value function $V^\pi(\mu)$,*

$$V^{\pi_1,\pi_2}(\mu) - V^{\pi'_1,\pi_2}(\mu) - V^{\pi_1,\pi'_2}(\mu) + V^{\pi'_1,\pi'_2}(\mu)$$
$$\leq \frac{2M\max_i |\mathcal{A}_i|}{(1-\gamma)^4} \sum_s d_\mu^{\pi'_1,\pi'_2}(s) \left( \|\pi_1(\cdot|s) - \pi'_1(\cdot|s)\|^2 + \|\pi_2(\cdot|s) - \pi'_2(\cdot|s)\|^2 \right)$$

We refer to [8] for complete proof for Lemmas C.1 and C.2. By using Lemmas C.1 and C.2, we can replace Lemma 3.5 in our paper with the following lemma.

**Lemma C.3.** *Given policy $\pi^k$ and marginalized advantage function $\bar{A}_i^{\pi^k}(s,a_i)$, for $\pi^{k+1}$ generated using independent NPG update, we have the following inequality,*

$$\Phi^{\pi^{k+1}}(\rho) - \Phi^{\pi^k}(\rho) \geq \left( \frac{1}{1-\gamma} - \frac{2M^3 \max_i |\mathcal{A}_i| n\eta}{(1-\gamma)^3} \right) \sum_s d_\rho^{\pi_i^{k+1},\pi_{-i}^k}(s) \sum_{i=1}^n \langle \pi_i^{k+1}(\cdot|s), \bar{A}_i^{\pi^k}(s,\cdot) \rangle.$$

The proof of Lemma C.3 closely follows proof of Lemma 3.5. First the potential function difference $\Phi^{\pi^{k+1}}(\rho) - \Phi^{\pi^k}(\rho)$ is decomposed following Lemma 3.5, with each term defined as a potential function that only differs by the local policy $\pi_i$. Different from 3.5, the decomposition is based on Definition C.1 instead. Then each term can be bounded using Lemma C.1 with the addition of Lemma C.2 to handle the residual terms.

Combining Lemma C.3 and Lemma B.3, we can get a similar convergence guarantee without the explicit definition of $\phi$, as stated in the following theorem:

**Theorem C.4.** *Consider a MPG in the sense of Definition C.1 with isolated stationary points, where the policy update follows NPG update in (7). For any $K \geq 1$, choosing $\eta = \frac{(1-\gamma)^2}{4nM^3 \max_i |\mathcal{A}_i|}$, we have*

$$\frac{1}{K} \sum_{k=0}^{K-1} NE\text{-}gap(\pi^k) \leq \frac{2M\phi_{max}}{K(1-\gamma)} (1 + \frac{8nM^3 \max_i |\mathcal{A}_i|}{c\delta^*(1-\gamma)} + \frac{K'}{2M}).$$

The proof of Theorem C.4 follows Theorem 3.6 and Corollary 3.6.1, and the only difference in the process is the different choice of $\eta$ from Lemma C.3. Compared to Corollary 3.6.1, Theorem C.4 incorporates additional terms on $M$ (distribution mismatch coefficient) and $|\mathcal{A}_i|$ (size of action space), but the $O(1/\epsilon)$ order is still maintained.

# D  Supporting Lemmas

**Lemma D.1.** *Given a reward vector $\mathbf{r} = [r_1, r_2, \cdots, r_n]$ where $r_1 > r_2 > r_3 > \cdots > r_n$ with weights $\pi_1 e^{\alpha r_1}, \pi_2 e^{\alpha r_2}, \cdots, \pi_n e^{\alpha r_n}$, we have*

$$\frac{\sum_{i=1}^n r_i \pi_i e^{\alpha r_i}}{\sum_{i=1}^n \pi_i e^{\alpha r_i}} \geq \frac{r_1 \pi_1 e^{\alpha r_1} + \sum_{i=2}^n r_i \pi_i e^{\alpha r_2}}{\pi_1 e^{\alpha r_1} + \sum_{i=2}^n \pi_i e^{\alpha r_2}}.$$

*Proof.* The above claim can be proved by induction. We first have

$$\frac{\sum_{i=1}^n r_i \pi_i e^{\alpha r_i}}{\sum_{i=1}^n \pi_i e^{\alpha r_i}} \geq \frac{\sum_{i=1}^{n-1} r_i \pi_i e^{\alpha r_i} + r_n \pi_n e^{\alpha r_{n-1}}}{\sum_{i=1}^{n-1} \pi_i e^{\alpha r_i} + \pi_n e^{\alpha r_{n-1}}}$$
$$= \frac{\sum_{i=1}^{n-2} r_i \pi_i e^{\alpha r_i} + \frac{r_{n-1}\pi_{n-1} + r_n\pi_n}{\pi_{n-1} + \pi_n}(\pi_{n-1} + \pi_n)e^{\alpha r_{n-1}}}{\sum_{i=1}^{n-2} \pi_i e^{\alpha r_i} + (\pi_{n-1} + \pi_n)e^{\alpha r_{n-1}}}.$$

This gives a new reward vector

$$r'_1 = r_1 > r'_2 = r_2 > \cdots > r'_{n-2} = r_{n-2} > r'_{n-1} = \frac{r_{n-1}\pi_{n-1} + r_n\pi_n}{\pi_{n-1} + \pi_n},$$

with weights

$$\pi_1 e^{\alpha r_1}, \pi_2 e^{\alpha r_2}, \cdots, \pi_{n-2} e^{\alpha r_{n-2}}, (\pi_{n-1} + \pi_n)e^{\alpha r_{n-1}}.$$

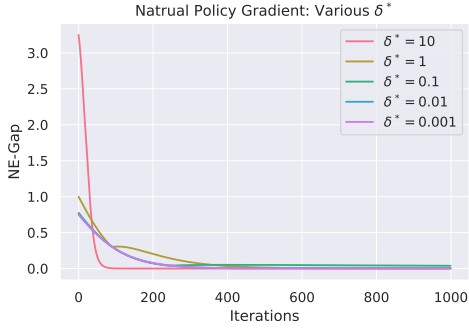
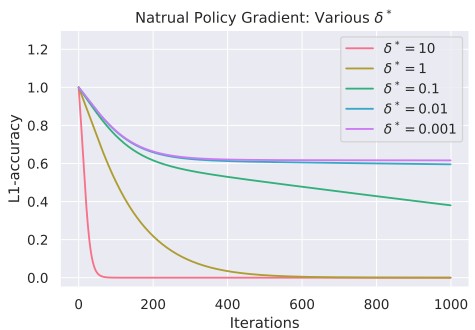

(a) NE-gap of matrix game with varying $\delta^*$     (b) $L_1$ accuracy of matrix game with varying $\delta^*$

Define $\pi'_{n-1} := \pi_{n-1} + \pi_n$ and by similar argument as above,

$$
\begin{aligned}
RHS &= \frac{\sum_{i=1}^{n-2} r_i \pi_i e^{\alpha r_i} + r'_{n-1} \pi'_{n-1} e^{\alpha r_{n-1}}}{\sum_{i=1}^{n-2} \pi_i e^{\alpha r_i} + \pi'_{n-1} e^{\alpha r_{n-1}}} \\
&\geq \frac{\sum_{i=1}^{n-3} r_i \pi_i e^{\alpha r_i} + \frac{r_{n-2} \pi_{n-2} + r'_{n-1} \pi'_{n-1}}{\pi_{n-2} + \pi'_{n-1}} (\pi_{n-2} + \pi'_{n-1}) e^{\alpha r_{n-2}}}{\sum_{i=1}^{n-3} \pi_i e^{\alpha r_i} + (\pi_{n-2} + \pi'_{n-1}) e^{\alpha r_{n-2}}} \\
&= \frac{\sum_{i=1}^{n-3} r_i \pi_i e^{\alpha r_i} + \frac{r_{n-2} \pi_{n-2} + r_{n-1} \pi_{n-1} + r_n \pi_n}{\pi_{n-2} + \pi_{n-1} + \pi_n} (\pi_{n-2} + \pi_{n-1} + \pi_n) e^{\alpha r_{n-2}}}{\sum_{i=1}^{n-3} \pi_i e^{\alpha r_i} + (\pi_{n-2} + \pi_{n-1} + \pi_n) e^{\alpha r_{n-2}}}.
\end{aligned}
$$

This gives a reward vector

$$
r''_1 = r_1 > r''_2 = r_2 > \cdots > r''_{n-2} = \frac{r_{n-2} \pi_{n-2} + r_{n-1} \pi_{n-1} + r_n \pi_n}{\pi_{n-2} + \pi_{n-1} + \pi_n},
$$

with weights

$$
\pi_1 e^{\alpha r_1}, \pi_2 e^{\alpha r_2}, \cdots, \pi_{n-3} e^{\alpha r_{n-3}}, (\pi_{n-2} + \pi_{n-1} + \pi_n) e^{\alpha r_{n-2}}.
$$

By induction, we have

$$
RHS \geq \frac{r_1 \pi_1 e^{\alpha r_1} + \sum_{i=2}^{n} r_i \pi_i e^{\alpha r_2}}{\pi_1 e^{\alpha r_1} + \sum_{i=2}^{n} \pi_i e^{\alpha r_2}}.
$$

$\square$

# E   Additional Numerical Experiments

We illustrate the impact of $\delta^*$ on the algorithm in the following example. We consider a 2-by-2 matrix game with the reward matrix

$$
r = \begin{bmatrix} 1 & 2 \\ 3 + \delta^* & 3 \end{bmatrix}.
$$

We run the NPG algorithm with the same initial policy under various values of $\delta^*$ ranging from $1e^{-3}$ to 10. The NE-gap and the $L_1$ accuracy over number of running steps are shown in Figures 2a-2b. As illustrated in the figure, $\delta^*$ plays an important role in the convergence of the algorithm. Larger $\delta^*$ results in faster convergence, which corroborates our theoretical findings.

