# OpenReview forum: "Provably Fast Convergence of Independent Natural Policy Gradient for Markov Potential Games"
_NeurIPS.cc/2023/Conference — NeurIPS 2023 poster_

### Official Review · Reviewer_XDXN · 2023-06-15

**Soundness:** 3 good
**Presentation:** 2 fair
**Contribution:** 2 fair
**Rating:** 6
**Confidence:** 5

**Summary:**

This work proposes an analysis of the independent natural policy gradient algorithm for Markov Potential games. Under technical assumptions on a sub-optimality gap problem dependent quantity and supposing access to exact (averaged) advantage functions, this paper provides a novel $O(1/\epsilon)$ iteration complexity to guarantee that the average Nash gap along iterations is smaller than the accuracy \epsilon, improving over the previously known $O(1/\epsilon^2)$ iteration complexity in the same independent learning setting. After discussing the potential game setting as a warm-up and generalizing to the MPG setting, the paper provides simulations in a synthetic potential game and a congestion game.


**Strengths:**

- The convergence analysis improves over the $O(1/\epsilon^2)$ iteration complexity in prior work under some technical assumptions.

- To the best of my knowledge, the analysis provided in this paper is new and the proofs seem to be correct (I went through the appendix, please see some additional comments below). However, please see some comments below regarding related works and quantities/notations to be precised for clarifications. In my opinion, the main technical novelty is Lemma 3.2 (and Lemma B.3 relying on the technical lemma C.1) which is key to obtain the result in Theorem 3.3. This result is interesting and connects the sum of the ‘gaps’ (over agents) for a fixed temperature parameter $\eta$ and for $\eta$ going to infinity.

- The paper is well-organized.


**Weaknesses:**

**1. About the suboptimality gap \delta_K and the $O(1/K)$ convergence rate**:
Theorem 3.3 provides an upper-bound on the average NE-gap which depends on the K-dependent quantity $\delta_K$ without further assumption, this provides a $O(1/K\delta_K)$ convergence rate as mentioned in the paper.
I think it should be clearly stated that the $O(1/K)$ convergence rate which is claimed in the abstract and in the contributions under ‘mild assumptions’ (as it is stated) is actually “asymptotic”.  Indeed, under assumption 3.2, the bound in Corollary 3.6.1 features an unknown constant (number of iterations) $K$’ that is not explicit in the problem parameters (such a constant is only guaranteed to exist under assumption 3.2). This is due to the fact that the NE gap is only controlled (independently of $\delta_k$) for a large enough number of iterations as it appears in l. 500 in the appendix (p. 17) under the chosen assumption.  The results that the paper compares to in Table 1 have stronger guarantees in the sense that they are not asymptotic. This is not clear in the presentation and the comparison to prior work.

**2. About Assumption 3.2**: this technical assumption guarantees that $\delta_K$ is uniformly lower bounded away from zero but it seems hard to interpret or give a meaning to this assumption although the ‘sub-optimality gap’ $\delta_k$ is a standard quantity in the bandit literature for instance. I am also not sure if this assumption is ‘mild’ as it is formulated. See the ‘Questions’ section for clarification questions regarding this assumption.

**3. Discussion of related works**: Some relevant related works are missing in the discussion.

**(a)** While Song et al. 2021 [27] is cited, it is not mentioned in the discussion that a $O(1/\epsilon)$ iteration complexity has been achieved in that work for MPGs with a Nash-coordinate ascent algorithm (see Section 5 Algorithm 7, note that sample complexity is given and the $O(1/\epsilon)$ iteration complexity appears in the proof when discarding the $O(1/\epsilon^2)$ sample complexity needed for policy evaluation). However, this algorithm is ‘turn-based’ and requires coordination and hence is not independent as the present work.

**(b)** Fox et al. 2022 derived an asymptotic convergence result for the independent natural policy gradient algorithm considered in this work that seemed to be later used by Zhang et al. 2022 [33] but this reference does not appear in related works. This same result seems also to be used in Proposition 3.1 of the paper.

**(c)** The results shown in this paper seem to have some similarities with the asymptotic convergence analysis provided by Khodadadian et al. 2022 in the single agent setting. For instance, the analysis in that paper introduces the optimal advantage function gap (see $\Delta$ as defined in Definition 2), a quantity similar to the sub-optimality gap $\delta_k$ in the present paper (up to the multi-agent setting). However, I would like to point out that this is just for the purpose of comparison and the present work has to overcome many difficulties related to the game theoretic setting and the multi-agent nature of the problem that make this work very different from Khodadadian et al. 2022. The abstract precises that the result improves over “ $O(1/\epsilon)$, that is achievable for the single-agent case”. Actually, Khodadadian et al. 2022 provide an asymptotic geometric convergence rate. Other recent related works even prove a global linear rate with increasing step sizes for the natural policy gradient algorithm (see for e.g. Xiao 2022, Section 4.2).

Fox et al. Independent natural policy gradient always converges in Markov potential games, AISTATS 2022.

Khodadadian et al. On linear and super-linear convergence of Natural Policy Gradient algorithm,  Systems and Control letters 2022.

Xiao. On the convergence rates of policy gradient methods. JMLR 2022

**(d)** minor remark: you might want to give reference to [Monderer and Shapley 1996, Potential Games, Games and Economic Behaviour 14, 124-143] which introduced this class of games in section 3.1 when you mention [12] (l. 128) which is much more recent.

**4. Regarding the definition of Markov potential games** (Definition 2.1), for a fair comparison in Table 1, it might be worth mentioning that this definition differs from the one considered in [8,16] although it matches the definition used in [33, 34]. Indeed, the potential function in Definition 2.1 is supposed to have a discounted cumulative structure whereas such a structure is not available in the more general definition considered in [8,16]. As a matter of fact, the analysis becomes more challenging in [8] for instance since showing the potential function improvement is then more involved in that case, another decomposition different from the decomposition in Lemma B.2 l. 440 is then used to guarantee policy improvement (see Lemma in [8]). Also the dependence with respect to some parameters such as $(1-\gamma)$ and the state action space sizes are usually improved with this additional structure.

**5. Originality of the analysis:** While Lemma 3.2 and its use is indeed new, the potential function improvement lemmas (Lemma 3.1, Lemma 3.5) and their proofs in appendix follow prior techniques used in [4, 33]. I suggest that authors mention this somewhere in the main part or in the appendix and emphasize the novelty of the analysis (Lemma 3.2). For instance Lemma A.1 was proved in [4], the proof of Lemma A.2 is almost identical to the proof in [4] while the proof of Lemma B.2 is very similar to the proofs in [33].

**6. Clarity**: Overall, writing can be substantially improved in my opinion. Few minor details below:

— l. 108: what do you mean by ‘multiple stationary points for the problem’? Stationary points if the potential function?

— l. 134-135: not very clear, see the ‘Questions’ section below.

— l. 136-137: ‘for any two sets of policies’. I guess you mean any two joint policies (in the product of the individual simplices) when you say sets.

— l. 158: ‘They are related by the following lemma’, the constants you just defined in l. 157 or the quantities defined few lines above in l. 153?

— notation $f(\infty)$ is used in Lemma 3.2 and does not seem to be defined before although it is quite obvious this corresponds to $\lim_{\alpha \to \infty} f(\alpha)$ as used in l. 153.

— Lemma A.4 in the appendix: the statement and the proof are not very precise. What is $\mathcal{R}_i$ (it seems only defined in the end of the proof of this lemma, or we can guess it from the title of the lemma)? I guess this is the set of reward functions which is known to have a particular structure for MPGs as you write it. $\mathcal{R}_i$ is a linear space in which ambient space? Please precise the proof even if we can guess the idea. Also, where is this result used?

—  Lemma C.1: if rewards are vectors, please precise somewhere in notations that inequalities hold for all the entries of the vectors.

**Typos (main part and Appendix)**:
 l. 91: $V_i(s)$; do you mean $V_i^{\pi}(s)$?

l. 476-477 (proof of Lemma B.2 in appendix): $\bar{A}_{f_i}^{\pi^k}$, what is $f_i$? I guess you mean $h_i$.

l. 479 to l. 480: $\pi_i^{k+1}(\cdot|s)$, $s$ is missing in the first term of the last equation and in the KL divergences.

l. 503 (proof of corollary 3.6.1): is there a missing $\phi_{\max}$ in front of $K’$ in the first inequality of the page (given l. 501)?

**Questions:**

1. The assumption of ‘isolated stationary policies’ seems to be needed to guarantee that $c>0$ as stated in Theorem 3.6 (as it is also stated and used in [33]). However this assumption is not stated in the theorem. How does the theorem guarantee that $c> 0$?

2. Please clarify the definition of ‘isolated stationary policies’. What do we mean by ‘stationary’ policies in this context? Although such a terminology is used in [33], it is not clear to the reader what this means, especially that ‘stationary’ has also another meaning for policies (time-independent). The reference Fox et al. 2022 above may help with this.

3. About assumption 3.2, isn’t it possible to use the assumption that the ‘stationary policies are isolated’ instead to obtain corollary 3.6.1? According to the comments l 248-250 in the paper (a short proof in the appendix may be useful then to state this implication properly), this assumption would guarantee that Assumption 3.2 hold. Moreover, the assumption that the ‘stationary policies are isolated’ seems to be already needed in Theorem 3.6 to guarantee that $c >0$. Would this mean that only the assumption that ‘stationary policies are isolated’ would be then sufficient for Theorem 3.6 to hold without needing Assumption 3.2?

4. Could you please precise the mathematical definition of $\pi_i^{*k}$? It seems to be only defined once with words in l. 134-135. Why should it be unique (‘the optimal solution …’)? Do you mean it is an optimal policy in the sense that it maximizes the averaged advantage function w.r.t. the policy of other agents but i being fixed to $\pi_{-i}^k$ (or the averaged reward function in the potential games setting)? This is also important to clarify the limit statement in l. 153.

5. l. 153: why do we have that $f(\eta) \geq 0$?


**Limitations:**

The paper mentions the MPG setting as a limitation in the conclusion.

**Extension to the stochastic setting:** While the analysis in the deterministic setting is an important step towards understanding the more practical stochastic setting where exact advantage functions are not available and can only be estimated via sampled trajectories, this analysis does not seem to be easily extendable to the aforementioned stochastic setting. This seems to be related to the fact that showing that the constant c is positive in the stochastic setting seems to be much harder if not hopeless even in a single agent setting (see for e.g. Mei et al. 2021). However, this is a limitation that also applies to prior work in MPGs analyzing independent natural policy gradient such as Zhang et al. 2022 [33]. A comment on this or an additional remark in the paper would be welcome. For instance, [8] analyzes the sample complexity in the stochastic setting but their algorithm does not cover the case where the regularization in the policy mirror descent-like update rule is not euclidean, KL regularization (which leads to natural policy gradient) is not covered.

   Mei et al. ‘Understanding the effect of stochasticity in policy optimization’ (Neurips 2021)

---

> ### Author Rebuttal · Authors · 2023-08-09
>
> We thank the reviewer for the detailed assessment and constructive feedback on the paper, with both main paper and appendix. We are encouraged by the fact that the reviewer finds our paper "well-organized" with "new and interesting analysis". Our submission has been revised to include previously missing relevant works, fix typos and rephrase potential obscurities raised by the reviewer. We also address the reviewer's concerns and questions below and sincerely hope that the reviewer would consider increasing the score.
>
> > About $\delta_K$ and rate... convergence is actually 'asymptotic'.
>
> A more detailed discussion on $\delta^k, \delta_K, \delta^*$ is provided **in Authors' Response to All**. We will clarify the "asymptotic" property of results in abstract, contributions, and Table 1 in our final version.
>
> > Asm. 3.2 guarantees... hard to interpret although... standard quantity in the bandit literature....
>
> The reviewer is correct that it is hard to guarantee a lower bound on $\delta^k, \forall k$. Therefore, we proposed the theoretical relaxation with Cor. 3.6.1 based on $\delta^*$. **Please see the details in Authors' Response to All.**
>
> In the bandit literature, the suboptimality gap indicates the structure of the environment. In the multi-agent setting, other agents can also be seen as part of environment w.r.t. a specific agent. Based on this intuition, we introduce this concept based on the marginalized reward and use Lem. 3.2 to capture the impact of $\delta^k$.
> > Related works
>
> We sincerely thank the reviewer for pointing out these previous works and we will include them in the final version.
> > Definition of MPGs
>
> Our formulation of MPG was originally adapted from that of [33, 34, R1, R2]. We consider this formulation to be well-known and somewhat standard in the literature. We will mention this difference in formulation with additional statements alongside Tab. 1 and Sec. 2.
> > Originality of the analysis
>
> We thank the reviewer for acknowledgement of the novelty of our analysis in Lem. 3.2, B.3, and C.1. The proofs of Lem. A.1 and A.2 are similar to [4], but we arrive at different conclusions due to new lemmas (Lem. 3.2 and B.3) as bridges. The analysis of Lem. B.2 is related to [33] but with some key differences. Firstly, we provide a new Lem. B.1 to better explain the final equality in l. 476, which doesn't appear in [33]. Secondly, we use Young's inequality in l. 479 to save a $\sqrt{n}$ factor compared with [33]. Nevertheless, we will make sure to mention these works while providing revised proofs of these lemmas.
> > Clarity
> - We use the definition of 'stationary point' from mathematics and optimization to denote a point with zero gradients. Incidentally, in our paper's context, the stationary point denotes a set of policies with zero policy gradients.
> - Yes, we used the 'set of policies' to denote a joint policy of product simplices. We will make sure to clarify at our first use of this specific term.
> - We meant the quantities defined in l. 153 and we will clarify it in the final version.
> - Yes, the definition of $f(\infty)$ will be included in our updated draft.
> - Yes, $\mathcal{R}\_i$ is the set of reward functions with a particular structure. The ambient space is $\mathbb{R}^{\prod\_{j=1}^n |\mathcal{A}\_j|}$. We wanted to provide a discussion over the structure of PGs in Lem. A.4 without further complicating other lemmas or theorems.
> - Here our intention was to show for a vector $\mathbf{r} = [r_1, ..., r_n]$, where $r_1 > r_2 ... > r_n$. We will make sure to clarify this in our statements.
>
> > Typos
>
> We thank the reviewer for the detailed evaluation of the paper. We will fix these typos in our final version.
> > 'isolated stationary policies’... as stated in Thm. 3.6 (and [33])... not stated in the theorem. How does the theorem guarantee that $c>0$?
>
> Yes, this assumption is required. We will add it to the final version.
> > What do we mean by ‘stationary’ policies in this context? Although such a terminology is used in [33], it is not clear... has also another meaning for policies (time-independent).
>
> A stationary policy is what our paper describes as a 'stationary point', a set of policies that has zero policy gradients. In this sense, 'isolated stationary policy' means no other stationary points exist in an open neighborhood of any stationary point. We will add this clarification in the revision.
> > ... isn’t it possible to use the assumption to obtain corollary 3.6.1? ... would guarantee that Assumption 3.2 hold.
>
> The mild Asm. 3.2 assumes the limit $\delta^*$ is larger than 0, which is required by Cor. 3.6.1. It is not possible to only use "stationary policies are isolated" to obtain Cor. 3.6.1.
>
> A counter example would be a 2-by-2 matrix game with $r_{11} = 1, r_{12} = 1, r_{21} = 2, r_{22} = 1$, where one NE would be $\pi_1 = (0, 1), \pi_2 = (1, 0)$. In this example, we have an isolated stationary policy with $\delta^* = 0.$
> > Precise definition of $\pi_i^{*k}$?
>
> The exact definition is that $\pi_i^{*k} \in \arg\max_{\pi_i} V_i^{\pi_i, \pi_{-i}^k}(\rho)$.
>
> > l. 153: why $f(\eta)\geq 0$?
>
> NPG updates $\pi_i^{k+1}$ as follows: $\pi_i^{k+1}(a_i|s) = \arg\max_{\pi_i} \eta \langle \bar{r}_i^k(\cdot), \pi_i(\cdot) \rangle - KL(\pi_i || \pi_i^k)$. Therefore, $f(\eta) = \sum_i \langle \bar{r}_i^k(\cdot), \pi_i^{k+1}(\cdot) - \pi_i^{k}(\cdot) \rangle \geq \sum_i KL(\pi_i^{k+1} || \pi_i^k) / \eta \geq 0$.
> > Extension to the stochastic setting:
>
> We follow [33] and only consider the oracle setting in our analysis. In general, an oracle can be estimated by Monte Carlo or temporal difference methods for the stochastic setting and can be analyzed similarly as [1]. However, as mentioned by the reviewer, it is much harder to handle $c$ in the stochastic setting. We will add the above comment and mention the sample-based results in [8] in our final version. We leave the related analysis as future work.
>
> *For brevity, please see response to reviewers Qt2L and tti4 for referenced works.*

---

> > ### Comment · Reviewer_XDXN · 2023-08-14
> > **post rebuttal**
> >
> > I confirm that I have read the authors’ rebuttal and I thank them for their response. I still think that assumption 3.2 is not a verifiable assumption under its current form. While the dependence on the suboptimality gap is natural as it appears in prior work (as I also mentioned in my original review and the authors further reconfirmed it), the suboptimality gap is usually **proved** to be a positive quantity (see for e.g. Lemma 3 in Khodadadian et al. 2022). Nevertheless, I acknowledge that this paper addresses a more challenging multi-agent setting. I raise my score to 6 following the authors’ rebuttal. Remaining limitations I see include Assumption 3.2, the asymptotic nature of the rate and the difficulty to extend the results to the stochastic setting given the dependence on the constant $c$ which positivity is difficult to guarantee in this setting (beyond assuming oracle access). I also encourage the authors to improve the writing, presentation and discussion of related works along the lines of the rebuttal.
> >
> > Khodadadian et al. 2022, On linear and super-linear convergence of natural policy gradient algorithm, Systems and Control Letters.

---

> > > ### Comment · Reviewer_Qt2L · 2023-08-14
> > >
> > > I am really sorry but I have taken ill and am in hospital so cannot engage any further with the review process.

---

> > > > ### Author Response · Authors · 2023-08-15
> > > >
> > > > We really appreciate your thoughtful comments and positive feedback in the original review. We hope you will recover soon.

---

> > > ### Author Response · Authors · 2023-08-15
> > >
> > > We thank the reviewer for carefully reading our rebuttal and raising the score. As observed by the reviewer, due to the challenging multi-agent setting, $\delta^k > 0$ is not necessarily true for all iterations as Lemma 3 in [R1]. For a few iterations $k$, it is possible to have $\delta^k = 0$. Therefore, we only define $\delta^* = \lim_{k \to \infty} \delta^k$ and use $\delta^*$ in the upper bound (cf. Table 1 and Corollary 3.6.1). We agree that the assumption about $\delta^*$ cannot be proved directly. Instead, it is an additional assumption on the structure of (Markov) potential games.
> > >
> > > We agree that the introduction of $c$ makes the analysis of stochastic setting difficult. This problem also exists in the previous works [33] and we will leave it as our future work. Additionally, we acknowledge the limitations and potential improvements to writing, presentation, and discussion of related works. All of them will be addressed in the final version.
> > >
> > > Moreover, our results can be extended to a more general form of potential function as in [8]. By using Lemmas 2 and 21 in [8], we can replace Lemma 3.5 in our paper with the following lemma.
> > >
> > > **Lemma B.4** Given policy $\pi^k$ and marginalized advantage function $\bar{A}_i^{\pi^k}(s, a_i)$, for $\pi^{k+1}$ generated using independent NPG updates, we have the following inequality,
> > >
> > > $$\Phi^{\pi^{k+1}}(\rho) - \Phi^{\pi^k}(\rho) \geq \left(\frac{1}{1-\gamma} - \frac{2M^3 \max_i |\mathcal{A}\_i| n \eta}{(1-\gamma)^3}\right) \sum_s d\_{\rho}^{\pi\_i^{k+1}, \pi\_{-i}^k}(s) \sum\_{i=1}^n \langle \pi\_i^{k+1}(\cdot|s), \bar{A}\_i^{\pi^k}(s, \cdot)\rangle. $$
> > >
> > > Using our new Lemma B.3 as a bridge, we can get a similar convergence guarantee without the explicit definition of $\phi$. Some additional terms about $M$ (distribution mismatch coefficient) and $|\mathcal{A}\_i|$ (size of action space) will be introduced, but the $O(1/\epsilon)$ order will be kept.
> > >
> > > [R1] Khodadadian et al. On linear and super-linear convergence of natural policy gradient algorithm, Systems and Control Letters, 2022.
> > >
> > > [33] Runyu Zhang, Jincheng Mei, Bo Dai, Dale Schuurmans, and Na Li. On the global convergence rates of decentralized softmax gradient play in markov potential games. Advances in Neural Information Processing Systems, 35:1923–1935, 2022.
> > >
> > > [8] Dongsheng Ding, Chen-Yu Wei, Kaiqing Zhang, and Mihailo Jovanovic. Independent policy gradient for large-scale markov potential games: Sharper rates, function approximation, and game-agnostic convergence. In International Conference on Machine Learning, pages 5166–5220. PMLR, 2022.

---

### Official Review · Reviewer_tti4 · 2023-06-29

**Soundness:** 3 good
**Presentation:** 3 good
**Contribution:** 3 good
**Rating:** 6
**Confidence:** 3

**Summary:**

The paper provides a new analysis for the (Markov) potential games with a $O(1/\epsilon)$ convergence rate. The new results are problem-dependent and may be a tighter guarantee for certain classes of potential games. Asymptotic guarantees are also provided to elaborate on the problem-dependent nature of the results.

**Strengths:**

The results it presents showcase an improvement over previous results on (markov) potential games. The new rates are now unaffected by the number of actions and are at the order of $O(1/\epsilon)$. The problem-dependent nature of the results may provide a tighter guarantee on a certain class of potential games and the new analysis methods may lead to future works. The results are extensively discussed and empirical results are provided to corroborates the theoretical results.

**Weaknesses:**

The new results do not seem to be directly comparable to the previous ones due to the use of a suboptimality gap. I encourage the authors to explicitly state this early in the paper to avoid confusion (e.g. in Table 1). I also encourage the authors to complement the work with more empirical results. For example, it would be interesting to see how the algorithm performs against previous ones when the suboptimal gap is very small. It may also be helpful to state when the suboptimality gap is very small, the results can degenerate into previous results.

**Questions:**

1. In Table 1, the results for PG + softmax [33] also include a c in the denominator. Is the c defined as the same as this paper?
2. It seems that in [33], the c in the denominator can be alleviated by using log barrier regularization. Will log barrier regularization have the same effect on the results presented in this paper?
3. In Definition 2.1, the potential function is assumed to take a specific form with $\phi$. However, this form does not seem to be needed in previous analyses. How important is this specific form to the analysis and can it be lifted?

**Limitations:**

Yes, it is discussed in the conclusion part.

---

> ### Author Rebuttal · Authors · 2023-08-09
>
> We thank the reviewer for the positive comments and suggestions concerning our paper. Please see our response below with respect to the specific comments and we sincerely hope that the reviewer would consider increasing the score.
>
>
> **Q1.** "The new results do not seem to be directly comparable to the previous ones due to the use of a suboptimality gap. I encourage the authors to explicitly state this early in the paper to avoid confusion."
>
>
> **Response:**
> We use Table 1 to provide a clear summary of our result and existing works, which performs the same function as Table 1 in [33]. $\delta^*$ was included in the iteration complexity in our Table 1.
> We will further clarify the use of the suboptimality gap in our statements in the abstract and contributions.
>
>
> **Q2.** "I also encourage the authors to complement the work with more empirical results."
>
>
> **Response:**
> We have added a purposefully constructed example to show the impact of $\delta^*$ on the algorithm in practice.
>
> We consider an example of a 2-by-2 matrix game with the reward matrix
> $r = \begin{bmatrix}
> 1&2\\\3+\delta^*&3
> \end{bmatrix}$.
>
> For the experiments, we have selected various values of $\delta^*$ ranging from $1e^{-3}$ to 10. We run the same algorithm with the same initial policy for all experiments, and plot both the NE-gap and the L1 accuracy (L1 distance between the current-iteration policies and Nash policies) of the algorithm. It can be seen from the experiments that $\delta^*$ indeed plays an important role in the convergence of the algorithm.
>
> **Please refer to newly attached pdf in "Authors' Response to All" for details.**
>
>
> **Q3.** "It may also be helpful to state when the suboptimality gap is very small, the results can degenerate into previous results."
>
>
> **Response:**
> In fact, the iteration complexity of independent NPG algorithms is the smaller of the results in this paper and those in [33]. The specific minimum value depends on $\delta^*$, which depends on the structure of (Markov) potential games.
>
>
>
> **Q4.** "Is the c defined as the same as this paper?"
>
>
> **Response:** Yes, the definition of $c$ is the same.
>
>
> **Q5.** "Will log barrier regularization have the same effect on the results presented in this paper?"
>
>
> **Response:** Since the log-barrier regularization repels the trajectory from regions with small policy values, we can derive a lower bound for policy value as Lemma 24 in [33]. While it is true that the log-barrier regularization can remove the dependence on $1/c$, the introduction of log-barrier parameter $\lambda$ makes the convergence rate slower by $O(1/\sqrt{K})$ since the upper bound of convergence rate has the form $\frac{c_1}{\lambda K} + c_2 \lambda$.
>
>
> **Q6.** "In Definition 2.1, the potential function is assumed to take a specific form with $\phi$. However, this form does not seem to be needed in previous analyses. How important is this specific form to the analysis and can it be lifted?"
>
>
> **Response:**
> Our formulation of MPG was originally adapted from that of [33, 34, R1, R2]. The definition of $\phi$ provides an additional structure to the problem, which changes the way of analysis and may lead to better convergence rates. However, it should be emphasized that even under this formulation, the best-known convergence rate is $O(1/\sqrt{K})$ [33, 34].
>
>
> [33] Runyu Zhang, Jincheng Mei, Bo Dai, Dale Schuurmans, and Na Li. On the global convergence rates of decentralized softmax gradient play in markov potential games. Advances in Neural Information Processing Systems, 35:1923–1935, 2022.
>
>
> [34] Runyu Zhang, Zhaolin Ren, and Na Li. Gradient play in multi-agent markov stochastic games: Stationary points and convergence. arXiv preprint arXiv:2106.00198, 2021.
>
>
> [R1] Macua, Sergio Valcarcel, Javier Zazo, and Santiago Zazo. "Learning Parametric Closed-Loop Policies for Markov Potential Games." International Conference on Learning Representations. 2018.
>
>
> [R2] Zazo, Santiago, et al. "Dynamic potential games with constraints: Fundamentals and applications in communications." IEEE Transactions on Signal Processing 64.14 (2016): 3806-3821.

---

> > ### Comment · Reviewer_tti4 · 2023-08-14
> >
> > Thank you for the explanation. I wonder if the results can be extended to the case with the more general form of potential function?

---

> > > ### Author Response · Authors · 2023-08-15
> > >
> > > We thank the reviewer for reading our rebuttal carefully. Yes, the results can be extended to a more general form of potential function as in [8]. By using Lemmas 2 and 21 in [8], we can replace Lemma 3.5 in our paper with the following lemma.
> > >
> > > **Lemma B.4** Given policy $\pi^k$ and marginalized advantage function $\bar{A}_i^{\pi^k}(s, a_i)$, for $\pi^{k+1}$ generated using independent NPG updates, we have the following inequality,
> > >
> > > $$\Phi^{\pi^{k+1}}(\rho) - \Phi^{\pi^k}(\rho) \geq \left(\frac{1}{1-\gamma} - \frac{2M^3 \max_i |\mathcal{A}\_i| n \eta}{(1-\gamma)^3}\right) \sum_s d\_{\rho}^{\pi\_i^{k+1}, \pi\_{-i}^k}(s) \sum\_{i=1}^n \langle \pi\_i^{k+1}(\cdot|s), \bar{A}\_i^{\pi^k}(s, \cdot)\rangle. $$
> > >
> > > Using our new Lemma B.3 as a bridge, we can get a similar convergence guarantee without the explicit definition of $\phi$. Some additional terms about $M$ (distribution mismatch coefficient) and $|\mathcal{A}\_i|$ (size of action space) will be introduced, but the $O(1/\epsilon)$ order will be kept.
> > >
> > > We thank the reviewer for the constructive suggestions. We will provide additional discussion and remarks on the structure of MPG in our revision. The detailed proof of this new Lemma will also be added to Appendix.

---

> > > > ### Comment · Reviewer_tti4 · 2023-08-15
> > > >
> > > > Thank you for your response. My concerns have been addressed and I still support accepting this work.

---

> > > > > ### Author Response · Authors · 2023-08-18
> > > > >
> > > > > We are glad to hear that your concerns have been addressed. Thank you again for your support and insightful comments.

---

### Official Review · Reviewer_Qt2L · 2023-07-05

**Soundness:** 4 excellent
**Presentation:** 4 excellent
**Contribution:** 2 fair
**Rating:** 7
**Confidence:** 4

**Summary:**

The article considers what the authors describe to be “natural policy gradient” learning in normal form and Markov potential games. This is the discrete time algorithm in which individual players' mixed strategies are multiplied by a softmax response to the opponent mixed strategies then normalised. Convergence rates are derived in both normal form and the Markovian potential games.

**Strengths:**

The article is nicely written, and the claimed results are supported by the theory.

The convergence rate results are nice, given the general difficulty to provide convergence rates in anything multiagent, and especially in Markovian games - of course restricting to situations in which there is a potential function for the full Markovian game makes things much less impossible, but it is still a hard problem.

The paper is nicely self-contained, and a real pleasure to read.

**Weaknesses:**

I have some doubts over the claimed results which I would like to see clarified.

The authors claim that Thm3. Implies a 1/eps convergence rate. I don’t buy it I’m afraid. c and delta_K are not controlled. They could be arbitrarily small, at least without further work. I suspect c is okay, although a sudden switch in best response could easily result in a very small pi_i^k(br_i(pi_{-i}^k)) coming into c late in the process. And since delta is the optimality gap when playing a mixed strategy, I find it very difficult to see how to constrain it effectively. (I think line 158 tells us that c is the smallest ever pi_i^k(br(pi_{-i}^k)), and delta_K is the smallest optimality gap that occurs up to time K – if I have misinterpreted this then my objections may dissipate!)

The synchronous form of "learning", and the tight construction of Markovian potential games with average reward, means the step up to Markovian settings is much less then in a less restrictive setting.

A more philosophical point is that the article assumes players can receive oracle information about the long term payoff of any action. While it makes for a nice compact paper, I think for NeurIPS the authors need to at least posit some suggestions for where learners might be able to access the advantage functions that are required to implement the method.

Furthermore, the dynamic is well known as the multiplicative weights algorithm, and I would expect the authors to compare their results with those presented under the multiplicative weights description.

**Questions:**

1. In (5), the function f has dependence only on alpha. However there is also important dependence on k, and on pi_{-i}^k. This may seem like pedantry, but I think it’s important (see below).
2. In lines 154/155 the definition of a^k_{i_q} is clumsy, due to possible multiplicities in the argmax for a^k_{i_p}. I think that a^i_q is only needed to define the optimality gap? If this is the case, why not just define delta^k directly max_a r^i(a) – max_{a\notin \argmax} r(a), or something like that? The current formulation taking a_i_q to be the max that is not a_i_p is incorrect, is noted to be incorrect in the text, but is used anyway.
3. In (6) we begin to get bitten by the lack of care in defining what the functions depend on. In particular, since the r_I^k functions depend on mixed strategies pi_{-i}^k, so does the delta term. And so delta_k may well be extremely small. Have I missed something here?
4. On line 159, we see that c_K and delta_K are the smallest such quantities that are observed in the first K iterations of the algorithm, and indeed that c is minimum c observed in the limit. The notation means that this path-dependency is somewhat obscured. The authors try to argue, later, that these quantities are not small. But I do not think the argument has been made strongly enough. Furthermore, it seems strange to take a limit of c_K values, but not take a limit of delta_K values in the theory that follows? This is clearly very relevant to my point made in "weaknesses".
5. First inequality in the proof of Theorem 3.3, I think should be an equality?
6. Line 259, I think we cannot claim that |delta^k-delta*| is small, since we are only assuming a lim inf?

**Limitations:**

No discussion of limitations is presented. I don't feel such a discussion would add much to the paper, but the review form asks the question!

---

> ### Author Rebuttal · Authors · 2023-08-09
>
> We thank the reviewer for the time and support of our paper as well as the valuable suggestions. We are encouraged by the fact that the reviewer finds our paper "nicely written", with "nice convergence rate results", andthat it is "self-contained". Please see our response below with respect to the specific comments.
>
> **Q1.** "$c$ and $\delta_K$ are not controlled. They could be arbitrarily small, at least without further work."
>
> **Response:** We understand the concern on $c$ and $\delta_K$ raised by the reviewer. **We refer to the above "Authors' Response to All" for a more detailed explanation.**
>
>
> **Q2.** "... the step up to Markovian settings is much less than in a less restrictive setting."
>
> **Response:** We agree that the Markov potential game (MPG) is a restrictive setting compared to the general multi-agent RL (MARL) setting. Considering the difficulty of multi-agent settings, we believe that the convergence analysis of independent NPG in MPG will be an important step in solving the general MARL problems.
>
> **Q3.** "A more philosophical point is that the article assumes players can receive oracle information. ... where learners might be able to access the advantage functions that are required to implement the method."
>
> **Response:** We follow [33] and only consider the oracle setting in our analysis. In general, an oracle can be estimated by Monte Carlo or temporal difference methods for the stochastic setting and can be analyzed similarly as [1]. However, as mentioned by the Reviewer XDXN, it is much harder to handle $c$ in the stochastic setting. We will add the above comment in our final version and leave the related analysis as future work.
>
>
> **Q4.** "... and I would expect the authors to compare their results with those presented under the multiplicative weights description."
>
>
> **Response:** As pointed out by [1] (after Lemma 15), NPG with softmax parameterization is "identical to the classical multiplicative weights updates".
>
>
> **Q5.** "function f ... also important dependence on k, and on $\pi_{-i}^k$."
>
>
> **Response:** At iteration $k$, both $\pi_i^k$ and $\bar{r}_i^k$ (oracle) are known and fixed. So the only decision variable for function $f$ is $\alpha$. For clarity, we will use $f^k(\alpha)$ in the final version.
>
>
> **Q6.** "the definition of $a^k_{i_q}$ is clumsy ... ."
>
> **Response:**
> The notations for action sets have been updated in the supplementary materials, please see **Authors’ Response to All** for the updated notations.
> For the main paper, it is enough to only define $\delta^k$ directly. The definitions of $a^k_{i_q}$ and $a^k_{i_p}$ were in fact used only for the proof of Lemma 3.2 in the appendix. We included them in our main paper only for consistency of analysis.
>
>
> **Q7.** "... we begin to get bitten by the lack of care in defining what the functions depend on"
>
>
> **Response:**
> Please refer to the response to Q5.
>
>
> **Q8.** "... Furthermore, it seems strange to take a limit of $c_K$ values, but not take a limit of $\delta_K$ values in the theory that follows"
>
>
> **Response:**
> $c^k > 0$ and $c > 0$ are guaranteed as shown in [33], but $\delta^k > 0$ is not necessarily true for all iterations. For a few iterations $k$, it is possible to have $\delta^k = 0$. Therefore, we only define $\delta^* = \lim_{k \to \infty} \delta^k$ and use $\delta^*$ in the upper bound (cf. Table 1 and Corollary 3.6.1).
>
>
> **Q9.** "First inequality in the proof of Theorem 3.3, I think should be an equality?"
>
>
> **Response:**
> It should be an inequality due to the fact that NE-gap takes the maximum over all agents, whereas the function $f$ is defined with respect to the summation.
>
>
> **Q10.** "Line 259, I think we cannot claim that $|\delta^k-\delta^*|$ is small, since we are only assuming a lim inf?"
>
>
> **Response:**
> The reviewer is correct. In fact, asymptotic convergence for this algorithm is guaranteed in previous works [33], and a limit exists. We will replace $\liminf$ by $\lim$ in the final version.
>
>
> [1] Alekh Agarwal, Sham M Kakade, Jason D Lee, and Gaurav Mahajan. On the theory of policy gradient methods: Optimality, approximation, and distribution shift. The Journal of Machine Learning Research, 22(1):4431–4506, 2021.
>
>
> [33] Runyu Zhang, Jincheng Mei, Bo Dai, Dale Schuurmans, and Na Li. On the global convergence rates of decentralized softmax gradient play in markov potential games. Advances in Neural Information Processing Systems, 35:1923–1935, 2022.

---

### Official Review · Reviewer_gegG · 2023-07-06

**Soundness:** 2 fair
**Presentation:** 2 fair
**Contribution:** 2 fair
**Rating:** 6
**Confidence:** 4

**Summary:**

This paper studies the convergence of Natural Policy Gradient method (NPG) in Markov Potential Game. Under stronger assumptions, the convergence result improves upon previous ones.

**Strengths:**

Under the considered setting, NPG is shown to achieve a $1/K$ convergence rate for multi-agent Markov Potential Game, matching the convergence rate in single-agent setting. Discussion on the parameters that appear in the convergence rate is presented in detail.

**Weaknesses:**

The convergence result only makes sense under the assumption that $c>0, \delta>0$. However, such assumption could be too strong and unnecessary (as the table 1 indicates).

**Questions:**

Line 177: "A small value for c generally describes a policy that is stuck at some regions far from an NE, yet the policy gradient is small. It has been shown in [17] that these ill-conditioned problems could take exponential time to solve even in single-agent settings."

In my opinion, this justification is not suitable. In single-agent settings, NPG has a nice convergence that does not depend on $1/c$, while [17] shows that PG can take exponential many iterations to converge. Thus, taking [17] as an excuse, it is okay to introduce certain assumptions (e.g. concentrability) to ensure better convergence, but directly assuming $c>0$ is still too strong (and possibly unnecessary).

---

> ### Author Rebuttal · Authors · 2023-08-09
>
> We thank the reviewer for the valuable feedback regarding the paper. Please see our responses below with respect to the specific comments. We believe that we have addressed all the concerns raised by the reviewer, and we sincerely hope that the reviewer would consider increasing the score.
>
> **Q1.** "$c > 0, \delta > 0$ could be too strong and unnecessary."
>
> **Response:** We understand the concern on $c$ and $\delta$ raised by the reviewer. **We refer to the above "Authors' Response to All" for a more detailed explanation.**
>
> **Q2.** "... this justification is not suitable. In single-agent settings, NPG has a nice convergence that does not depend on $1/c$, while [17] shows that PG can take exponential many iterations to converge ... directly assuming $c>0$ is still too strong (and possibly unnecessary)."
>
> **Response:**
> In single-agent RL, policy gradient depends on a product of the advantage function, the occupancy measure, and the action probability (Eqn. 10 in [1]). Therefore, it is possible for PG algorithm to make a small update although the advantage function is significant. Single-agent NPG solved this issue using the Moore-Penrose inverse of Fisher information matrix to cancel out occupancy measure as well as action probability. However, in MARL, Fisher information matrix does not fully cancel out everything, since its calculation only uses local policy. Based on this observation, we make a comparison between single-agent PG and multi-agent NPG.
>
> Note that we are not the first work to introduce $1/c$ in the analysis of multi-agent independent NPG algorithms. The same assumption is made in [33] with similar accompanying statements saying that "Based on our analysis and numerical results, even for natural gradient play—which is known to enjoy dimension-free convergence in single agent learning we find in the multiagent setting that it can still become stuck in these undesirable regions. Such evidence suggests that preconditioning according to the Fisher information matrix is not sufficient to ensure fast convergence in multi-agent learning. "
>
> We will rephrase our statements accordingly in the final version.
>
> [1] Alekh Agarwal, Sham M Kakade, Jason D Lee, and Gaurav Mahajan. On the theory of policy gradient methods: Optimality, approximation, and distribution shift. The Journal of Machine Learning Research, 22(1):4431–4506, 2021.
>
> [33] Runyu Zhang, Jincheng Mei, Bo Dai, Dale Schuurmans, and Na Li. On the global convergence rates of decentralized softmax gradient play in markov potential games. Advances in Neural Information Processing Systems, 35:1923–1935, 2022.

---

> > ### Comment · Reviewer_gegG · 2023-08-18
> >
> > Thanks for the detailed response. I now accept the justification of introducing $c, \delta$ in analyzing the complexity rate, and I will raise my rate accordingly. By the way, perhaps it would be better to highlight that the $O(1/K)$ convergence is in some sense asymptotic under such conditions.

---

> > > ### Author Response · Authors · 2023-08-18
> > >
> > > We thank the reviewer for carefully reading our rebuttal and raising the score. We will clarify the "asymptotic" property of results in the abstract, contributions, and Table 1 in our final version.

---

### Author Rebuttal · Authors · 2023-08-09


## Authors' Response to All (Discussion on $c$ and $\delta^*$):

We wholeheartedly thank all the reviewers for their time and their constructive feedback on our paper. The reviewers' comments provided us with great insights on how to increase clarity and reduce potential confusion about our paper. All the reviewers have raised similar concerns regarding our assumptions on $c > 0$ and $\delta^* > 0$ in the paper. Please note that reference numbers and line numbers are based on the supplementary material (MPG Main\&Appendix.pdf).

First, for simplicity, we recall the definitions of $c^k, c\_K, c$ and $\delta^k, \delta\_K, \delta^*$ in the potential game setting (Lines 154-157 in “MPG Main\&Appendix.pdf”). For agent $i$ at iteration $k$, define $a\_{i\_p}^k \in \arg\max\_{a\_j \in \mathcal{A}\_i} \bar{r}\_i^k(a\_{j}) =: \mathcal{A}\_{i\_p}^k$ and $a\_{i\_q}^k \in \arg\max\_{a\_j \in \mathcal{A}\_i\backslash \mathcal{A}\_{i\_p}^k} \bar{r}\_i^k(a\_{j})$, where $\mathcal{A}\_{i\_p}^k$ denotes the set of the best possible actions for agent $i$ in the current state and $\bar{r}\_i(a\_i) = \mathbb{E}\_{a\_{-i} \sim \pi\_{-i}}[r\_i(a\_i, a\_{-i})]$. We define

$$c^k := \min\_{i\in [n]} \sum\_{a\_j \in \mathcal{A}\_{i\_p}^k} \pi\_i^k(a\_j) \in (0, 1),\quad
\delta^k  := \min\_{i \in [n]} [\bar{r}\_i^k(a\_{i\_p}^k) - \bar{r}\_i^k(a\_{i\_q}^k)] \in (0, 1). $$

Additionally, we denote
$c_K := \min_{ k\in[K]} c^k; c := \inf_{K \to \infty} c_K; \delta_K := \min_{k\in[K]}\delta^k, \delta^* := \lim_{k \to \infty} \delta^k$.

- The effect of $c$ was discussed in our submission at l. 175-183. Our exact definition of c is indeed the same as the $c$ defined in [33], with similar motivation and justifications. Combining Lemma 2 in [33] and Proposition 3.1 in this paper, $c^k$ asymptotically converges to 1. Since $c^k > 0$ for any softmax policy parameterization ($\pi_{i}^k(a_i|s) > 0$) and $c^k$ asymptotically converges to 1, we have $c := \inf_k c^k > 0$. This can also be observed in Fig. 1a in the original paper. Therefore, we believe our assumption on $c$ is mild and has been used within the same context in prior work.

- The introduction of the suboptimality gap $\delta^k$ enables us to draw a crucial connection between the difference in the potential function and the NE-gap using Lemma 3.2. As stated in Section 3.3, the best rate without this assumption is $O({1/\sqrt{K}})$.

	We would also like to justify our introduction of the suboptimality gap with reference to two separate lines of work. In single-agent RL, Khodadadian et al. [R1] proved asymptotic geometric convergence with the introduction of "optimal advantage function gap $\Delta^k$". $\Delta^k$ is very similar to our definition of $\delta^k$. Additionally, the notion of suboptimality gap, though different in formulation, is commonly used in the multi-armed bandit literature [R2]. In both works, the introduction of a suboptimality gap greatly benefits the analysis. It also serves a similar purpose in our work.

	We agree with the reviewers that requiring $\delta^k > 0$ to hold for all iteration $k$ is indeed restrictive. However, we found that occasional $\delta^k \approx 0$ does not affect the global convergence rate. Motivated by this observation, we have proposed a theoretical relaxation with Cor. 3.6.1. Instead of requiring the suboptimality gap to be always non-zero, we only require the limit (existence of a limit is due to asymptotic convergence guarantee in [33]) to be non-zero (Assumption 3.2), which is far from restrictive. Recalling our definition in Eq. (6), the suboptimality gap is defined as the performance gap between the best and second-best classes of actions. Having a zero suboptimality gap implies that all actions of one agent belong to one class with the exact same expected reward values, which implies a zero NE-gap for the specific client.

Moreover, we provide more experiments regarding special scenarios in MPGs in the attached pdf file as requested by Reviewer tti4.

We will revise and rephrase the related statements in order to reduce the potential confusion to readers in the final version. Additionally, we will incorporate valuable suggestions concerning other related literature, and correct the typos and notation errors pointed out by various reviewers.

[33] Runyu Zhang, Jincheng Mei, Bo Dai, Dale Schuurmans, and Na Li. On the global convergence rates of decentralized softmax gradient play in markov potential games. Advances in Neural Information Processing Systems, 35:1923–1935, 2022.

[R1] Khodadadian, Sajad, et al. "On linear and super-linear convergence of Natural Policy Gradient algorithm." Systems \& Control Letters 164 (2022): 105214.

[R2] Lattimore, T. and Szepesvári, C., 2020. Bandit algorithms. Cambridge University Press.

---

### Decision · Program_Chairs · 2023-09-21

**Decision:**

Accept (poster)

**Comment:**

The paper established fast convergence of independent natural policy gradient for MARL in Markov potential games. The technical contribution is solid and has been unanimously acknowledged by the reviews (and I concur). Please make sure to address all the comments in preparing the final version of the paper.